# MaAsLin 3: refining and extending generalized multivariable linear models for meta-omic association discovery

William A. Nickols[1,2], Thomas Kuntz[1,2], Jiaxian Shen [1,3,4], Sagun Maharjan[2], Himel Mallick [5,6,7], Eric A. Franzosa [1,2,8], Kelsey N. Thompson [1,2,8,10], Jacob T. Nearing [1,2,8,10] & Curtis Huttenhower [1,2,8,9,10] ✉

Microbial community analysis typically involves determining which microbial features are associated with properties such as environmental or health phenotypes. This task is impeded by data characteristics, including sparsity (technical or biological) and compositionality. Here we introduce MaAsLin 3 (microbiome multivariable associations with linear models) to simultaneously identify both abundance and prevalence relationships in microbiome studies with modern, potentially complex designs. MaAsLin 3 can newly account for compositionality either experimentally (for example, quantitative PCR or spike-ins) or computationally, and it expands the range of testable biological hypotheses and covariate types. On a variety of synthetic and real datasets, MaAsLin 3 outperformed state-of-the-art differential abundance methods, and when applied to the Inflammatory Bowel Disease Multi-omics Database, MaAsLin 3 corroborated previously reported associations, identifying 77% with feature prevalence rather than abundance. In summary, MaAsLin 3 enables researchers to identify microbiome associations more accurately and specifically, especially in complex datasets.

Quantitative microbial community analyses often aim to identify community features (for example, microorganisms, genes or metabolites) associated with metadata (for example, human health outcomes, diet or environmental conditions). Various procedures have been developed to test for these associations, but there is still relatively little consensus on how to best approach differential abundance (DA) testing in microbiome studies[1]. In large part, this is driven by the unusual properties of microbiome data, including compositionality, sparsity and heteroskedasticity[2–5]. Like most omics data, they also generally present challenges due to their inherent high dimensionality[6], requiring efficient algorithms and corrections for multiple hypothesis testing.

In response to these challenges, a variety of methods for identifying differentially abundant features have been developed for microbial community profiles. Although none is perfect, three widely used examples that produce consistent and accurate benchmarking results are ALDEx2 (analysis of variance (ANOVA)-like differential expression), ANCOM-BC and MaAsLin 2 (refs. [1,7,8]). To address compositionality in microbiome data, ALDEx2 updates a Bayesian Dirichlet prior based

[1]Department of Biostatistics, Harvard T.H. Chan School of Public Health, Boston, MA, USA. [2]Harvard Chan Microbiome in Public Health Center, Harvard T.H. Chan School of Public Health, Boston, MA, USA. [3]Division of Gastroenterology, Massachusetts General Hospital and Harvard Medical School, Boston, MA, USA. [4]Clinical and Translational Epidemiology Unit, Massachusetts General Hospital and Harvard Medical School, Boston, MA, USA. [5]Division of Biostatistics, Department of Population Health Sciences, Weill Cornell Medicine, Cornell University, New York, NY, USA. [6]Department of Statistics and Data Science, Cornell University, Ithaca, NY, USA. [7]Division of Gastroenterology and Hepatology, Weill Cornell Medicine, Cornell University, New York, NY, USA. [8]Infectious Disease and Microbiome Program, Broad Institute of MIT and Harvard, Cambridge, MA, USA. [9]Department of Immunology and Infectious Diseases, Harvard T.H. Chan School of Public Health, Boston, MA, USA. [10]These authors contributed equally: Kelsey N. Thompson, Jacob T. Nearing, Curtis Huttenhower. ✉e-mail: chutten@hsph.harvard.edu

on the observed read counts, performs a centered log-ratio transformation on samples from the posterior and runs a significance test on these posterior ratios[9]. In a different strategy, ANCOM-BC (analysis of composition of microbiomes with bias correction)[10] (and subsequently ANCOM-BC2)[11] uses an expectation–maximization procedure to estimate the compositionality-induced bias when comparing abundances between multiple groups; it then tests for differences in the groups after accounting for this bias. Finally, MaAsLin 2 fits linear models on log-transformed relative abundances, as its evaluations found this to be comparably accurate to explicitly accounting for compositionality while preserving flexibility and numerical stability[12]. Furthermore, several evaluations have found that simple models like MaAsLin's often outperform more complicated DA methods[8,13,14].

These methods also address zero inflation using various strategies, but typically only assess feature prevalence insofar as it contributes to abundance associations (see 'Prevalence modeling in prior tools' in the Supplementary Information). However, microbial presence alone can be phenotypically important: pathogens may cause disease at very low abundance[15,16], certain taxa are more frequently detected in patients with colorectal cancer[17], rare taxa may signal shared communities from transplants or contamination[18], and some taxa are restricted to specific geographic or demographic groups[19–21]. Identifying phenotypes linked specifically to presence or absence offers clear scientific value[8,22].

To bridge these gaps, MaAsLin 3 (1) explicitly identifies both prevalence and abundance associations; (2) addresses compositionality via a median coefficient comparison or by using data from absolute abundance protocols when available; and (3) extends inference to new covariate types. It supports both count and relative abundance inputs across taxonomic, functional and metatranscriptomic data. In simulations, MaAsLin 3 improved precision by up to 0.27 over MaAsLin 2 while maintaining similar recall, and it performed favorably compared to other DA methods. When applied to the HMP2 Inflammatory Bowel Disease Multi-omics (IBDMDB) cohort[23], it replicated known associations but revealed that 77% of associations were prevalence based. The MaAsLin 3 software, documentation, tutorials and datasets are freely available at https://huttenhower.sph.harvard.edu/maaslin3/.

## Results

Here, we present MaAsLin 3, which carries out microbiome feature DA testing while accounting for sparsity, compositionality, high dimensionality, complex study designs, absolute abundance protocols and phenotypes influenced by feature presence or absence. It models presence/absence via logistic regression and nonzero abundance via log-linear models, addressing compositionality by comparing abundance coefficients against their median. MaAsLin 3 takes as input a per-sample feature table and metadata table and returns a table of associations with summary and per-association plots. Compared to MaAsLin 2, MaAsLin 3 distinguishes between prevalence and abundance associations, accounts for compositionality with experimental or computational techniques, and supports new covariate types, enabling detection of more specific biological signals with broader applicability to complex experimental designs.

MaAsLin 3's algorithm consists of (1) normalizing microbial community feature abundance profiles (by default, total-sum scaling to obtain relative abundances), (2) generating a parallel prevalence (present (1) versus absent (0)) profile and retaining a separate subset of data with nonzero abundances, (3) log transforming (base 2) the abundance dataset to stabilize variance, (4) performing a modified logistic regression on the prevalence dataset and linear regression on the abundance dataset, and (5) combining the quantitative effects into an overall effect for each feature–metadatum pair (Fig. 1a and Methods). MaAsLin 3 is, therefore, classified as a hurdle model with a distribution for presence/absence and a distribution for present values, like MAST or scREHurdle for single-cell RNA sequencing[24,25]. However, it differs in key ways from other zero-inflated models for microbiome

data such as metagenomeSeq[26] that do not model prevalence associations with covariates.

Separating prevalence (presence/absence) from abundance (quantity when present) is important both technically and biologically. Technically, many common transformations (for example, log) require replacing zeros with pseudo-counts, but results can be sensitive to the pseudo-count choice[11,27,28]. Biologically, presence and abundance can reflect distinct phenomena (for example, infection versus commensalism) and may even cancel when aggregated. For example, in the IBDMDB cohort, regressing *Eubacterium rectale* against age using pseudo-counts shows no association ($\beta = 0.001$, 95% confidence interval (CI) −0.009 to 0.011, $P = 0.86$; Fig. 1b), but separating the effects reveals significant, opposing trends: a negative abundance association (linear $\beta = -0.015$, 95% CI −0.020 to −0.010, $P = 1 \times 10^{-9}$) and a positive prevalence association (logistic $\beta = 0.010$, 95% CI 0.003 to 0.017, $P = 0.005$). This suggests *E. rectale* is more likely to be present with age but at lower abundance when present.

In addition to prevalence modeling, another major improvement of MaAsLin 3 is the option to identify associations either on a true (experimental) absolute abundance scale or by including an additional (inferred) component accounting for compositionality. When only testing for relative abundance associations, MaAsLin 3 simply fits the specified models on relative abundance data. Biologically, both relative and absolute abundance phenotypes are plausible, but more sophisticated methods are necessary to provide an option for also inferring potential absolute-scale associations[2–4]. To identify such associations, MaAsLin 3 can also test for a difference between each feature's coefficient and the median of these coefficients (Methods). When fewer than half of the community's features are changing in absolute abundance—a frequent and biologically plausible assumption in microbiome DA testing[27,29,30]—this modified test can be interpreted as a test against the null hypothesis that an association's coefficient is zero on an absolute scale[29] (Extended Data Fig. 1a and Methods). However, even when this assumption is violated, the ranking of the coefficients on the relative and absolute scales will be identical in the absence of sparsity and will likely be similar even with sparsity[3]. Therefore, this median comparison can be interpreted as a test for whether a particular feature's association with a covariate is different from the typical association between features and the covariate on an absolute scale. As shown subsequently, the associated test results align closely with true absolute abundance associations derived from experimental data.

### MaAsLin 3 outperforms other DA methods

To evaluate how MaAsLin 3's prevalence modeling and compositionality correction perform in data with known associations, synthetic microbiome profiles were simulated from a modified zero-inflated log-normal distribution using SparseDOSSA 2 (ref. 5). SparseDOSSA 2 generates an underlying null distribution of features based on real metagenomic data, adds in abundance (log-linear) and prevalence (logistic) associations on a simulated absolute scale and samples reads to produce resulting counts and normalized relative abundances. Therefore, the simulated data available for DA testing are on the relative scale, but the underlying associations are known on an absolute scale. The resulting data were analyzed with MaAsLin 3, ALDEx2, ANCOM-BC2 and MaAsLin 2 (all using predominantly default parameters; Methods). The RNA DA methods DESeq2 (ref. 31) and edgeR[32] were also evaluated since they have historically been used for microbiome analysis, although previous analyses have found them to not be well suited for microbiome data[1,12,33,34]. For clarity, unless otherwise stated, comparisons to the RNA methods as well as to MaAsLin 3 with spike-in information and ALDEx2 with scale information are included only in the Supplementary Information. This was because—consistent with previous analyses—the RNA methods performed poorly on microbiome data, and the scale-informed models incorporate information not available to the other tools, preventing a fair comparison.

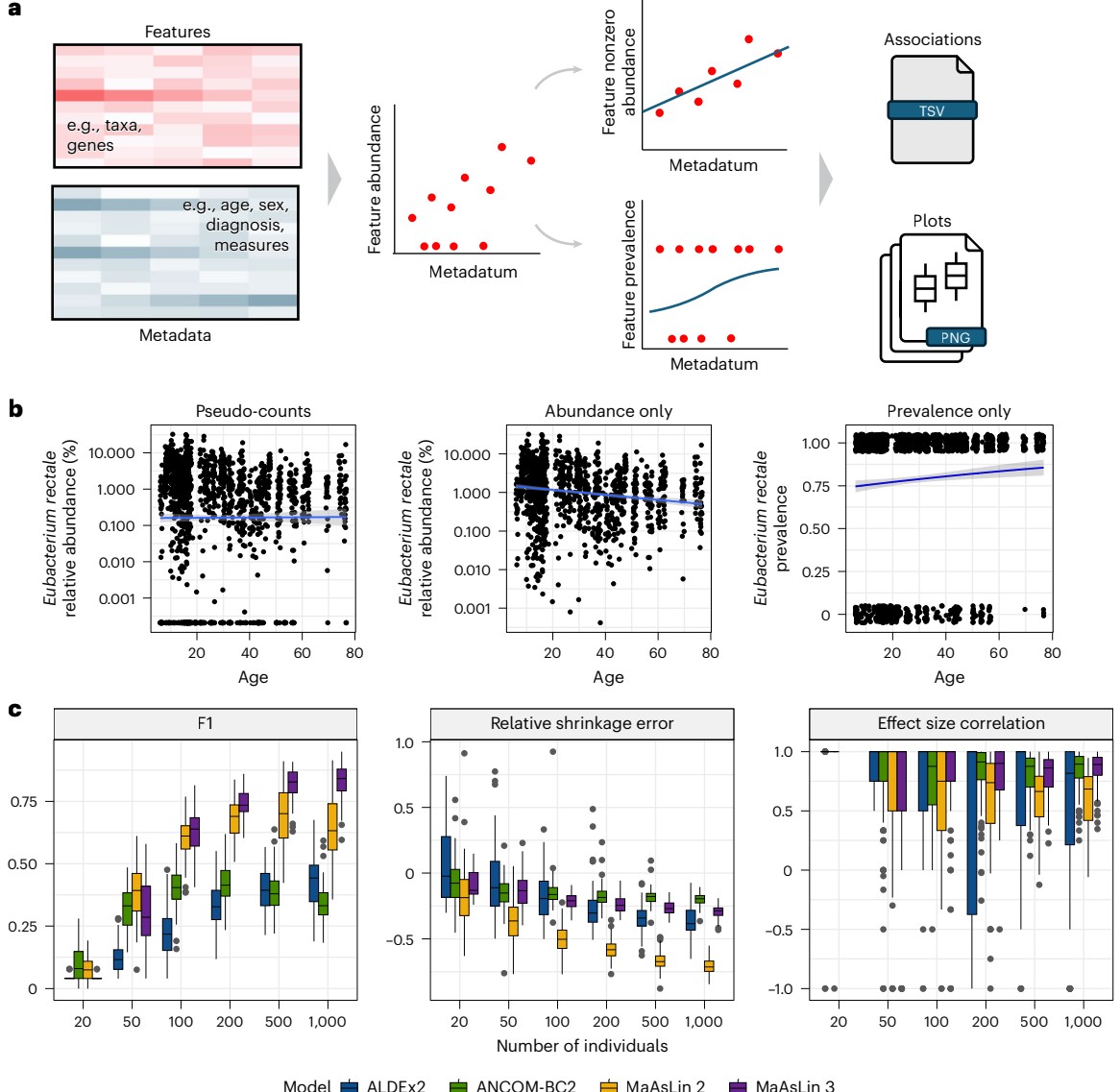

**Fig. 1 | MaAsLin 3 enables both abundance and prevalence modeling with improved accuracy. a**, MaAsLin 3 model overview. MaAsLin 3 takes as input a table of microbial community feature abundances, as counts or relative abundances, and a corresponding set of metadata (phenotypes, covariates, exposures, and so on). These feature data are normalized, filtered, split into prevalence ans log-transformed nonzero abundances, and fit with a modified logistic model and a linear model, respectively. A table of associations is produced indicating the summary statistics corresponding with each feature–metadatum association. **b**, Using all metagenomes from the HMP2 IBDMDB cohort[23], *E. rectale* shows no association with age when zeros are replaced with pseudo-counts, but it shows a negative nonzero abundance association and a positive prevalence association. The blue line represents the fitted model, with the 95% CI in gray **c**, MaAsLin 3 outperforms other DA methods in simulations. MaAsLin 3 and other common DA methods were run on 100 synthetic log-normal datasets from SparseDOSSA 2 (ref. 5) across six different sample sizes represented by the number of individuals within each simulation. For these simulations, 100 features and 5 metadata were simulated with 10% of the feature–metadatum pairs having true associations with coefficients sampled

uniformly from 2.5 to 5, half of which were positive and half of which were negative. Half of the true associations were abundance associations; the rest were prevalence associations. The read depth per sample was drawn from a log-normal distribution with a mean of 50,000 (analogous to the number of informative reads per dataset, such as amplicon sequencing). Significant associations (no model-fitting errors, $q$ value < 0.1, joint $q$ value for MaAsLin 3) were considered correct if they matched the true associations in the feature and metadatum. A mismatch in association type—abundance versus prevalence—was allowed for all methods since no method besides MaAsLin 3 reports association type. F1 is the harmonic mean of precision and recall; 1 is optimal. The relative shrinkage error is the difference between the absolute fit and true coefficients divided by the true coefficient, averaged over the significant associations; 0 is optimal. The effect size correlation is the Spearman correlation between the fit and true coefficients per metadatum averaged over the metadata; 1 is optimal. Each point represents a simulated dataset. Box plots display the median, interquartile range, and whiskers extending to the most extreme values within 1.5 times the interquartile range (IQR) and individual points indicating outliers.

When comparing modeled-versus-true associations, MaAsLin 3 obtained a median F1 score as high or higher than any other method on datasets containing more than 50 samples (Fig. 1c). MaAsLin 3 and ALDEx2 maintained the highest precision at all sample sizes (average precision ≥ 0.82 and ≥ 0.99), although the precision of most methods

decreased with larger sample sizes due to increasing power to discover spurious compositionality-induced associations (Extended Data Fig. 2). Offsetting its high precision, the recall of ALDEx2 never rose above 0.27 for any sample size, whereas MaAsLin 3's recall rose to 0.85 by 1,000 samples. When just considering true prevalence associations

in well-powered 1,000-sample settings, even previous methods that did not explicitly model prevalence achieved substantial recall (0.51 for ANCOM-BC2 and 0.93 for MaAsLin 2). This reinforces that other methods identify prevalence associations as abundance associations since they can be considered a subtype of abundance association. At small sample sizes, the recall of MaAsLin 3 was lower (average 0.18 with 50 samples) than of ANCOM-BC2 (0.22) or MaAsLin 2 (0.25) due to logistic models typically requiring more samples to reach significance and MaAsLin 3's linear model component only considering nonzero values (reducing effective sample size). However, because statistically significant associations are often targets for time-intensive follow-up analysis, precision is typically more important than recall, and low-precision methodologies can raise concerns about reproducibility[33]. In these low sample settings, MaAsLin 3 had higher precision (average 0.99 at 50 samples) than the methods with higher recall (0.89 for MaAsLin 2 and 0.65 for ANCOM-BC2). Regarding the fit effect sizes, coefficients were biased toward zero in models that rely heavily on pseudo-counts, but the ordering of the effect sizes was generally accurate for all methods (Fig. 1c, and see 'Effect size accuracy' in the Supplementary Information).

Next, we evaluated the sensitivity of MaAsLin 3 to assumptions about how zeros are introduced into the dataset. By default, SparseDOSSA 2 uses a zero-inflated model, so zeros represent both biological (true) zeros and sequencing zeros (below the limit of detection). This potentially impedes methods that use pseudo-counts, which effectively assume all zeros are sequencing zeros. When SparseDOSSA 2 was modified to produce only sequencing zeros, the precision of MaAsLin 3 decreased modestly from 0.97 to 0.82, recall and coefficient correlations improved for all tools, and coefficient bias was mostly unaffected (Extended Data Fig. 4). However, with only sequencing zeros, zeros constituted 39% of the taxonomic abundance table, much less than when biological zeros were also included (77%). The proportion of zeros found in the real datasets subsequently analyzed (66%, 80% and 97%) was more consistent with a model involving both sequencing and biological zeros, but even when simulating only technical zeros, MaAsLin 3 maintained higher precision than any other tool with similar recall.

All methods produced similar results when varying the proportion of associations that were related to abundance rather than prevalence, except that the recall of the MaAsLin tools decreased with fewer prevalence associations (Extended Data Fig. 5). Still, MaAsLin 3 produced a higher recall than any other tool that controlled the false discovery rate (FDR), regardless of the proportion of associations that were due to prevalence. When read depth was correlated with a metadatum of interest, including read depth as a covariate prevented precision loss, although all methods exhibited decreased recall with this correction (as expected; Extended Data Fig. 6).

When multiple observations per participant were available, precision declined across ALDEx2, MaAsLin 2 and MaAsLin 3 due to increased power to detect spurious compositional effects. MaAsLin 3 showed a larger drop in precision (0.95 with no repeats versus 0.73 with ten samples per individual) than ALDEx2 (1.00 versus 0.99), but it achieved substantially higher recall (0.80 versus 0.27 at 10 samples). Most of MaAsLin 3's precision loss came from small, compositionally induced prevalence effects: 70% of its false discoveries had prevalence coefficients with absolute values < 1 at ten repeats. Applying a post hoc threshold of 1 to all model coefficients (Extended Data Fig. 7b) restored MaAsLin 3's precision (≥0.92), aligning it with ALDEx2 and ANCOM-BC2, without impacting recall. In contrast to all these results, ANCOM-BC2's precision improved with repeats (0.55 with none versus 0.96 with 2), but its recall stayed low (≤0.33 for any repeat count; Extended Data Fig. 7b). All coefficients correlated well with their true values, with MaAsLin 3 and ANCOM-BC2 outperforming others (0.82 and 0.85 versus 0.68 for ALDEx2 and 0.56 for MaAsLin 2 at ten repeats). These results suggest MaAsLin 3 effectively handles longitudinal data, and its precision can

be further optimized by thresholding coefficients (for example, at 20% of the maximum reliable value).

**MaAsLin 3 is robust to violations of modeling assumptions.** To test MaAsLin 3's robustness beyond its model assumptions, we applied the ANCOM-BC simulation framework[10], introducing abundance effects and structural zeros, where entire groups lacked specific features (Methods). While these simulations are extreme in sparsity and compositionality, MaAsLin 3, ANCOM-BC2 and ALDEx2 showed high precision at low sample sizes (for example, at 50 samples: 0.96, 0.95 and 1.00, respectively), although precision for MaAsLin 3 and ANCOM-BC2 worsened at 1,000 samples (to 0.64 and 0.68; Extended Data Fig. 8). At any sample size, the precision of MaAsLin 2 never exceeded 0.49. Recall was high (>0.91) for all methods except ALDEx2, which traded recall (0.54 at 20 samples to 0.97 at 1,000 samples) for precision. All methods' fitted effect sizes correlated well with the true values (correlation ≥ 0.73), with MaAsLin 3 and ANCOM-BC2 yielding slight shrinkage (−12%, −13%), ALDEx2 more shrinkage (−31%) and MaAsLin 2 inflation (+7.9%) at 1,000 samples. Separately, we applied a metadata randomization protocol[1] to 38 real datasets (100 randomizations each) and MaAsLin 3 maintained a low FDR: <1% of joint $q$ values were significant per run, with 90% of runs yielding none (Extended Data Fig. 9). Only 9 of 3,800 replications yielded false positive prevalence hits. In contrast, on unshuffled data, MaAsLin 3 identified significant associations in most datasets: 2.3% of abundance, 3.3% of prevalence and 6.5% of overall associations were significant. These findings mirror prior results for ALDEx2, ANCOM-II and MaAsLin 2 (ref. 1) and suggest that prevalence-based associations may be more common than abundance associations in real data, for biological (for example, niche specialization or individual personalization), statistical (for example, only associations large enough to involve feature loss can be detected) or technical (for example, sequencing depth restricts detecting abundance changes) reasons.

## MaAsLin 3 improves accuracy beyond simpler regression frameworks

To identify which components of MaAsLin 3's modeling strategy improved accuracy, we separately evaluated abundance and prevalence association accuracy. In both a 100-sample typical scenario and a 1,000-sample high-power scenario designed to expose flaws in the model, the default median abundance comparison improved precision at any fixed recall level (Fig. 2a). To test the inference of associations from only relative abundances against a model incorporating absolute abundance information, we modified SparseDOSSA 2 to mimic a spike-in experiment in which a known quantity of a reference feature is added to each sample and used for normalization (Methods). Running MaAsLin 3 on this spike-in data further improved precision, although at 1,000 samples, the default MaAsLin 3 outperformed the spike-in model (average precision 0.84 versus 0.73; Extended Data Fig. 10a), particularly when few true associations existed. This is partly due to the default model incorporating median uncertainty in its significance test, which effectively filters out false positives with small effect sizes. This was balanced by spike-in normalization yielding slightly higher recall (0.83 versus 0.75).

For prevalence modeling, two features of MaAsLin 3 substantially improved precision. First, MaAsLin 3 performs a data augmentation step for prevalence modeling to help address cases of linear separability. For each sample, two additional data points with small weights are added: one with the same covariates and the feature set to present, and one with the same covariates and the feature set to absent, a procedure equivalent to the Bayesian Diaconis–Ylvisaker prior that extends directly to mixed-effects models (Methods)[35,36]. This augmentation procedure improves precision over the equivalent simple logistic regression and prevents spurious effects that would result from just a few samples with influential covariates having present or absent taxa (Fig. 2b). Second, MaAsLin 3 compares the prevalence coefficient for

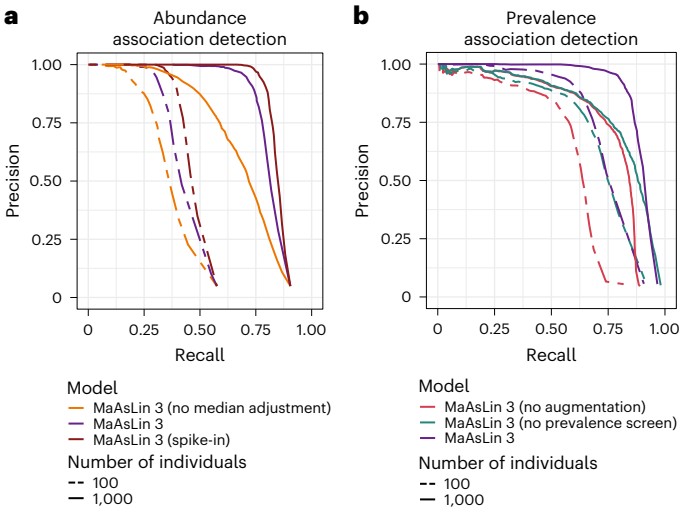

**Fig. 2 | MaAsLin 3's default model components improve accuracy beyond simpler regression models. a**, Precision versus recall across a range of *q*-value thresholds for various MaAsLin 3 abundance modeling options: without a median adjustment for compositionality when only using relative abundance data; MaAsLin 3's default settings (with the median adjustment when using relative abundance data); and without the median adjustment using data from a (simulated) experimental spike-in procedure. The versions were run on the same 100 synthetic log-normal datasets from SparseDOSSA 2 as in Fig. 1c. Unlike Fig. 1, significant associations (no model-fitting errors, individual *q* value < 0.1) were only considered correct if they matched the true associations in the feature, metadatum and type of association (prevalence/abundance). **b**, Precision versus recall for MaAsLin 3 prevalence modeling options: without data augmentation to account for separability but with prevalence coefficient screening; without prevalence coefficient screening (allowing any significant prevalence associations) but with data augmentation; and MaAsLin 3's default setting (with both augmentation and prevalence coefficient screening). The same datasets as in **a** were used. Curves farther to the right are better.

an association to the corresponding abundance coefficient and flags prevalence effects likely induced by abundance changes (Methods). Because sequencing depth is not infinite, when a feature's abundance is reduced biologically with a covariate, its likelihood of not being detected increases, a phenomenon that can manifest as a prevalence effect. Discarding prevalence associations flagged as likely abundance induced also improved precision while holding recall fixed (Fig. 2b). Thus, MaAsLin 3's new default features improve accuracy when detecting both abundance and prevalence associations.

To determine the source of the remaining error, mainly precision loss with large sample sizes, we increased the read depth from 50,000 to 50,000,000 reads (see 'Remaining inaccuracies are attributable to finite read depth' in the Supplementary Information). This eliminated almost all inaccuracies in the MaAsLin 3 models, although the improvements were not as consistent or substantial for other DA methods (Extended Data Fig. 10).

### MaAsLin 3 approximates absolute associations using relative data

Until now, all simulation results assumed that only a small proportion (10%) of the feature–metadatum associations were non-null, an assumption common in DA methods[27,29]. However, this assumption can be violated in practice if most features in the community are changing in abundance. To demonstrate that no method reliant on only relative abundances is exempt from this assumption, SparseDOSSA 2 was used to create datasets in which increasing proportions of the features were positively associated with the metadata on the absolute scale. According to compositional data theory[3], the relative abundance coefficients may be biased from the absolute abundance coefficients by a

per-metadatum additive shift, but they should correlate perfectly (or, given sparsity, at least correlate well). As expected, the inferred coefficients from all relative abundance models were negatively biased. This bias became more extreme as more features were positively associated with the metadata (Fig. 3a). ANCOM-BC2 and MaAsLin 3 were most robust to the assumption violation, but even they were negatively biased at any level and very negatively biased (average bias −2.8 and −3.1) when most features were associated with the covariates. As expected, only MaAsLin 3 and ALDEx2 with spike-in (that is, experimental) information were fully robust to the assumption violation. By contrast, the correlation between the true abundance effects and the estimated effects was mostly unaffected by the proportion of true associations for all methods. Despite these biases, ALDEx2 (no scale information) and MaAsLin 3 (with and without spike-ins) maintained high precision regardless of the proportion of features that were differentially abundant. All tools except ALDEx2 maintained similar recall regardless of the proportion of features changing, and ALDEx2 with no scale information had much lower recall while ALDEx2 with scale information had much higher recall when a large proportion of features were changing.

To evaluate how well these conclusions hold in real data, we reprocessed three datasets from diverse communities with absolute abundances experimentally quantified via spike-ins, digital PCR or flow cytometry: an infant gut dataset[37] (*n* = 178 infants, 650 samples), a mouse diet dataset[38] (*n* = 12 mice, 45 samples) and an inflammatory bowel disease and primary sclerosing cholangitis (IBD/PSC) dataset[39] (*n* = 170 participants, 1 sample per participant). Since relative and absolute abundance coefficients only correlate perfectly in the absence of sparsity, these datasets were also chosen for their range of sparsity: the infant dataset contained 97% zeros in its data matrix compared to 66% in the mouse diet dataset and 80% in the IBD/PSC dataset. For each dataset, MaAsLin 3 was run on both the relative and experimentally estimated absolute abundance data, and the corresponding coefficients were compared. As expected, in the sparse infant gut dataset, the relative abundance and experimentally estimated absolute abundance coefficients correlated the least (Spearman correlation 0.47), and the majority of the experimentally determined absolute abundance coefficients were positive (that is, the taxon increased with age) while a majority of the relative abundance coefficients were negative (Fig. 3b). In the mouse diet dataset, the relative and experimentally estimated absolute coefficients were well correlated for both the time variable and the diet variable. However, while the relative and absolute abundance coefficients for time were approximately equal, the diet coefficients showed a clear offset, consistent with theoretical expectations that relative abundance coefficients are correlated with, but possibly shifted from, the corresponding absolute abundance coefficients. In the IBD/PSC dataset, all relative abundance coefficients were well correlated with their experimentally estimated absolute abundance counterparts, and no systematic shift was observed for any coefficient.

Next, the relative abundance coefficients produced by each method were compared to the experimentally estimated absolute abundance coefficients from MaAsLin 3. Except for in cases of heavy sparsity, a regression of the relative abundance coefficients on the absolute abundance coefficients should give a slope of 1 and a high correlation. Indeed, for all datasets except the infant gut cohort, MaAsLin 3 and ANCOM-BC2 achieved these results (Fig. 3c). By contrast, both ALDEx2 and MaAsLin 2—heavily reliant on pseudo-counts that can bias the abundance relationship—produced coefficients with much weaker correlations (average correlation for non-infant datasets 0.52 and 0.59 versus 0.95 and 0.96 for ANCOM-BC2 and MaAsLin 3) and more attenuated slopes (average slope for non-infant datasets 0.46 and 0.45 versus 0.97 and 0.98). Typically, the infant gut cohort produced both the weakest correlations and the most attenuated slopes, suggesting that the assumption that would have yielded high correlations—independence between the absolute abundance of the

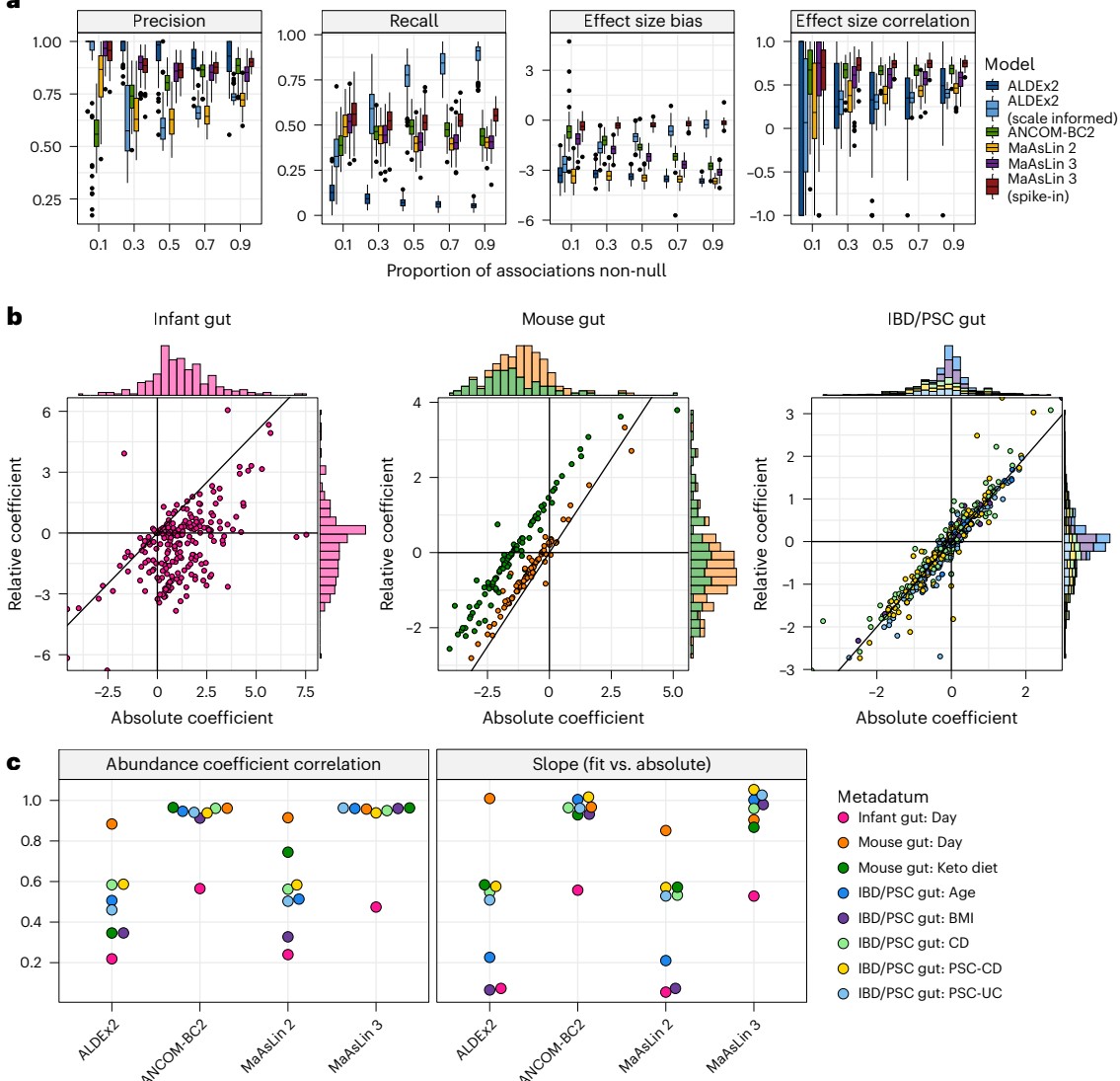

**Fig. 3 | Properties of absolute abundance data that are identifiable on the relative scale are well identified by MaAsLin 3. a,** All methods show increasing bias but little change in precision or coefficient correlations when relying on relative abundance data in which more features have true positive associations. MaAsLin 3 and other common DA methods were run on 100 synthetic log-normal datasets from SparseDOSSA 2 generated as in Fig. 1c but with 90% of the associations positive and the sample number fixed at 100. The precision is the proportion of assigned associations with *q* values below 0.1 that were true associations. The recall is the proportion of true associations that were identified by the DA tools with *q* values below 0.1. The effect size bias is the mean of the fit coefficients minus their true coefficients for true associations. The effect size correlation is the Spearman correlation between the fit and true coefficients per metadatum averaged over the metadata. 1 is optimal for all metrics except the effect size bias, for which 0 is optimal. Each point represents a simulated dataset. Box plots display the median, interquartile range and whiskers extending to the most extreme values within 1.5 times the IQR and individual points indicating outliers. **b,** Relative and experimentally estimated absolute abundance

coefficients agree to varying degrees on three real datasets with experimentally determined (spike-in, digital PCR or flow cytometry) absolute abundances. MaAsLin 3 was run on both the experimental absolute abundances and on the corresponding relative abundances, and the corresponding coefficients are plotted against each other with one point per feature–metadatum pair. **c,** MaAsLin 3 and ANCOM-BC2 relative abundance regressions best agree with the experimental absolute abundance regressions. ALDEx2, ANCOM-BC2, MaAsLin 2 and MaAsLin 3 were run on the relative abundances and compared to the experimentally determined absolute abundance associations from MaAsLin 3. For each method, for each metadatum in each dataset, the Spearman correlation between the fit relative abundance coefficients and the experimental absolute abundance coefficients was computed over all features. Similarly, for each method, for each metadatum in each dataset, the per-feature relative abundance coefficients were regressed on the per-feature experimentally estimated absolute abundance coefficient. A correlation of 1 and a slope of 1 are optimal. BMI, body mass index.

total community and the particular taxa present (see 'With sparsity' in the Supplementary Information)—was severely violated. Indeed, the infant gut is a low-biomass environment, and *Klebsiella* and *Escherichia* blooms were reported in this cohort, reinforcing that the total community absolute abundance depended on the taxa present[18,37]. This highlights the importance of using absolute abundance experimental protocols when the community's total abundance is expected to shift dramatically, since both assumptions enabling absolute abundance

inference from relative abundance data—half of the features remaining unchanged and independence between the total absolute abundance and the particular taxa present—are violated.

## MaAsLin 3's linear model extensions enable new experimental designs
In addition to differentiating prevalence from abundance associations, MaAsLin 3 enables five new types of inference (Table 1): specifying

general mixed-effects models, testing omnibus differences among three or more levels, testing level-versus-level differences in ordered predictors, testing contrasts among fit coefficients, and controlling for feature-specific covariates (particularly for metatranscriptomics experiments; Table 1, Methods, 'MaAsLin 3's linear model extensions enable new experimental designs' in the Supplementary Information and Supplementary Fig. 1b).

In particular, MaAsLin 3's feature-specific covariate testing directly implements our recommended best practices for differential expression testing in any metatranscriptomics analysis[40]. To demonstrate this, we applied these improvements to bioBakery 3 profiles from the HMP2 IBDMDB metatranscriptomes[23,41] (Supplementary Fig. 2, Methods and see 'HMP2 results' in the Supplementary Information). Of the 209 significant diagnosis or dysbiosis associations identified (no model-fitting errors, $q$ value < 0.1), 142 (68%) were abundance associations, suggesting that while pathway expression varies in both abundance and prevalence, it predominantly differs in abundance (once controlled for functional potential). To additionally demonstrate MaAsLin 3's group and ordered covariate testing, all IBDMDB metagenomes from participants with Crohn's disease (CD; $n$ = 750) were used to regress species abundance and prevalence (from both MetaPhlAn 4 (ref. 42; recent) and MetaPhlAn 3 (ref. 41) profiles) on food consumption frequency data. Consistent with the group and ordered covariate models testing similar but distinct hypotheses, 59 of the 132 significant ordered associations had a corresponding group-wise association, and 59 of the 107 significant group-wise associations had a corresponding significant ordered association (MetaPhlAn 4). Of the 797 diet associations discovered in either MetaPhlAn version, only 28 overlapped between the two, but among the associations that did overlap, there was high agreement in the magnitude of the association (Supplementary Fig. 3 and see 'HMP2 results' in the Supplementary Information).

## MaAsLin 3 refines biomarker discovery in inflammatory bowel diseases

To further explore associations in real data, we next applied MaAsLin 3 to the IBDMDB cohort to assess taxa associated with CD and ulcerative colitis (UC)[23]. Since the biomarkers of IBD are well established, we aimed to reconstruct generally accepted prior associations while assessing consistency across taxonomic classifiers and DA testing methods. MetaPhlAn 3 (as previously published)[23] and 4 (updated) were used to construct taxonomic profiles (750 CD samples from 65 individuals, 459 UC samples from 38 individuals, and 428 non-IBD samples from 27 individuals), and the resulting profiles were tested for associations with disease status and dysbiosis while controlling for age, antibiotic usage, read depth and repeated sampling (Methods).

Extending a previous analysis[23], we stratified the IBDMDB cohort into pediatric (age < 16) and adult (≥16) groups and analyzed differential microbial associations with IBD. Among 60 significant positive associations (FDR $q$ value < 0.1, $|\beta| > 1$ and no model-fitting errors), only two overlapped between age groups: *Citrobacter freundii* (SGB10083) and *Klebsiella pneumoniae* (SGB10115), both with CD dysbiosis[43–45]. The adult group had 35 unique positive associations including *Enterocloster* species (for example *E. bolteae*, SGB4758), *Clostridia* like *Flavonifractor plautii* (SGB15132) and inflammation-linked taxa such as *Hungatella hathewayi* (SGB4741)[46,47] (Fig. 4a). UC was associated with fewer taxa (6 versus 27 in CD), with enrichment of *Blautia faecicola* (SGB4867) and *Blautia obeum* (SGB4811). Pediatric-specific associations (23 total) included known IBD-linked microorganisms like *Escherichia coli* (SGB10068) and *Ruminococcus gnavus* (SGB4584), along with probiotic species such as *Bifidobacterium breve* (SGB17247) and *Lactobacillus acidophilus* (SGB7044)[48–50] (Fig. 4b). Negative associations were more consistent across age groups: 53 of 223 overlapped, especially losses of *Faecalibacterium prausnitzii* species genome bins (SGB15316, SGB15318, SGB15332, SGB15342), *Phocaeicola vulgatus*

**Table 1 | Linear model extensions enable new experimental designs**

| Feature | Explanation | Example use case |
|---|---|---|
| General mixed-model specification | Any valid lme4 (ref. 70) formula can be specified for the relationship between abundance or prevalence and covariates. | In a study with two treatments and two populations, an interaction term can be specified to evaluate the difference in the treatment's effect on species abundances between the populations. |
| Omnibus testing | For a covariate with multiple levels (categories), this is an ANOVA-style test of whether the different levels have significantly different coefficients. | In a study with participants from multiple countries, this could test whether species abundances differ among countries without testing for any individual pairwise differences. |
| Level-versus-level differences | For a covariate with ordered levels, this is a test for whether the coefficients for the differences between consecutive levels are significantly nonzero. | In a study of participants with different stages of colorectal cancer, this could test whether species abundances differ between consecutive stages. |
| Contrast testing | This is a test of whether a linear combination of coefficients is significantly different from a constant. When the linear combination is one coefficient minus another and the constant is zero, this tests for a difference in the coefficients. | In a study with samples from different soil types, this could test whether species abundances differ between each pair of soil types. |
| Feature-specific covariates | For a covariate that should be different for each feature, this includes the relevant covariate in each feature's model. | In a metatranscriptomic study with paired metagenomic data, this controls for a gene's DNA abundance when regressing that gene's RNA abundance to identify transcriptional rather than copy number differences. |

(SGB1814), *E. rectale* (SGB4933), *Bacteroides uniformis* (SGB1836) and multiple *Alistipes* and *Blautia* species, confirming broad depletion of commensals during IBD[51–54].

Analysis of MetaPhlAn 4 profiles using MaAsLin 3 revealed 372 significant associations, with 77% (287) related to prevalence and 89% (254) of those being negative, reinforcing that IBD largely alters microbial presence, not just abundance[55,56]. Notably, *Dysosmobacter welbionis* (SGB15078) was significantly reduced in prevalence in both CD and UC dysbiosis (prevalence $\beta_{UC}$ = −4.14 (95% CI$_{unadjusted}$: −5.86, −2.42),

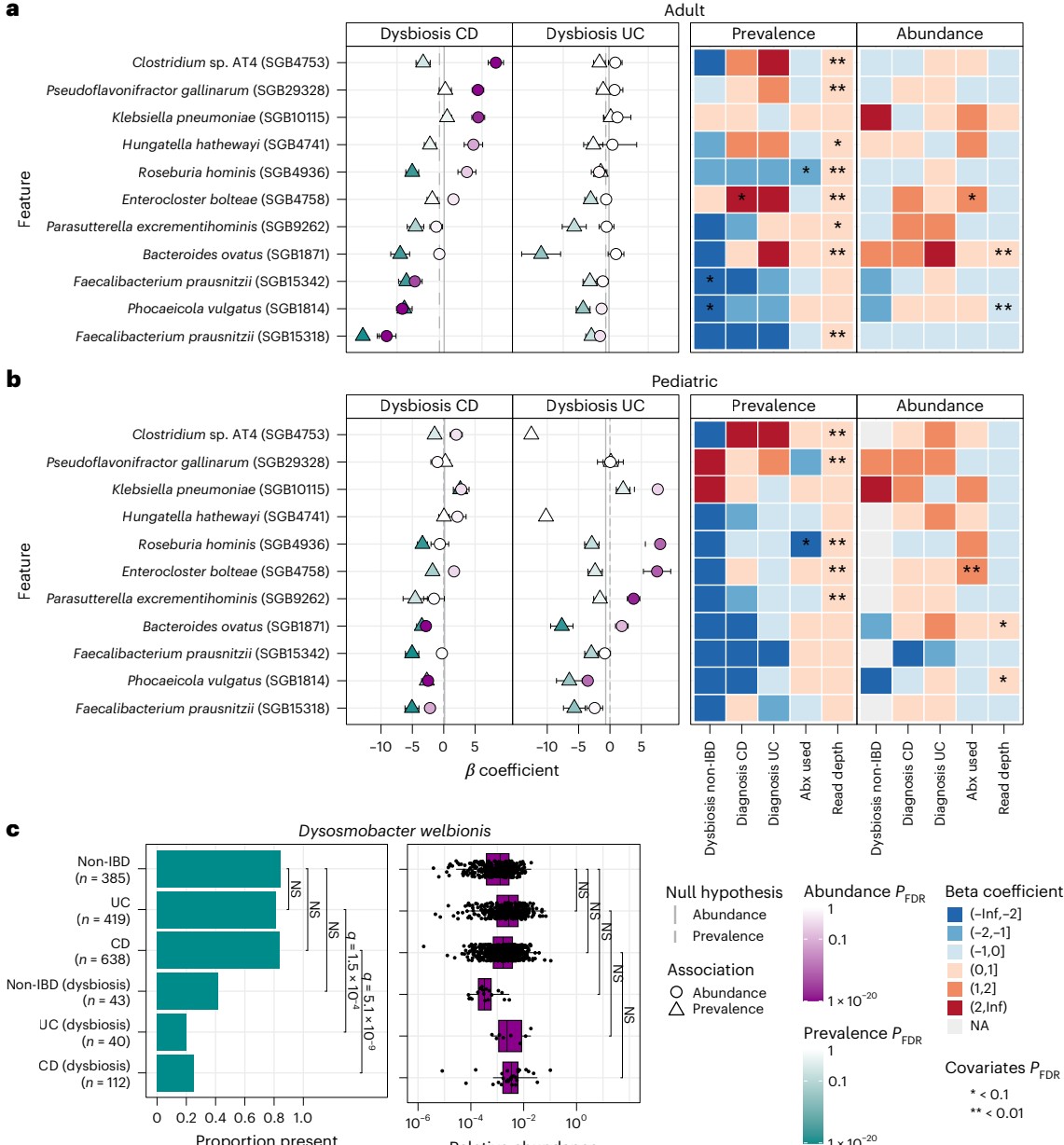

**Fig. 4 | MaAsLin 3 applied to the HMP2 IBDMDB verifies and extends previous gut microbiome associations with IBD. a,b,** The species-level abundances from the HMP2 cohort as determined by MetaPhlAn 4 were regressed in MaAsLin 3 using a model equivalent to that previously published[23] incorporating disease-stratified dysbiosis, disease diagnosis, antibiotic usage, read depth and a per-participant random intercept in individuals at least 16 years of age (*n* = 1,078, across 82 individuals) (**a**) or under 16 (*n* = 541, across 42 individuals) (**b**). Both **a** and **b** show a default MaAsLin 3 output summary figure that has been subset to highlight species associated with either adult or pediatric dysbiosis. The estimated coefficients and their standard errors are represented by points and bars on the left. **c,** *D. welbionis* prevalence differed between dysbiosis and non-dysbiosis, while abundance did not differ. Bars show the comparisons evaluated in the MaAsLin 3 model (two-sided test) after controlling for the aforementioned covariates and FDR correcting over all associations. Each point represents a simulated dataset. Box plots display the median, interquartile range, and whiskers extending to the most extreme values within 1.5 times the IQR and individual points indicating outliers. NA, not applicable. NS, not significant.

$\beta_{CD}$ = −3.46 (−4.47, −2.46); *q* values = 1.5 × 10⁻⁴, 5.1 × 10⁻⁹), although its abundance when present was not significantly reduced (abundance $\beta_{UC}$ = −1.12 (−2.18, −0.06), $\beta_{CD}$ = 0.15 (−0.64, 0.95); *q* values = 0.24, 1), a pattern that MaAsLin 2 failed to distinguish (Fig. 4c). Depending on the direction of causality, this may suggest that the organism's phenotypic effects are exerted when it is present regardless of abundance, or that it is so sensitive to an inflamed gut that it is driven below the limit of detection during IBD.

When the other DA methods were applied to the MetaPhlAn 4 profiles, most associations identified by MaAsLin 3 overlapped with those of another method (83% were also identified by MaAsLin 2, 32% by ALDEx2 and 4% by ANCOM-BC2; Supplementary Fig. 4). This overlap suggests MaAsLin 3's results are robust and driven by true biological phenomena rather than variation in methodology. By contrast, among ALDEx2's associations, 64% were unique to ALDEx2, as 177 of its 253 discovered IBD associations involved diagnosis, not dysbiosis (versus 35 of the 244 for MaAsLin 3). Among ALDEx2's dysbiosis associations, 74 of the 76 were also discovered by MaAsLin 3. Consistent with MaAsLin 3's detection of *D. welbionis* loss in UC and CD dysbiosis, MaAsLin 2 and ALDEx2 both estimated significant depletions of *D. welbionis*, although without distinguishing abundance from prevalence. ANCOM-BC2 flagged only 11 associations as significant in the entire dataset, all of

which overlapped with MaAsLin 3. These results support the plausibility of the new detail shed on the IBD gut microbiome by MaAsLin 3's richer and more nuanced DA model components.

## Discussion

DA testing is a key component in most microbial community analyses, but ambiguity remains regarding the best approaches[1]. In large part, this is driven by the challenging properties of microbiome data, particularly compositionality, sparsity and high dimensionality. To address these, we introduced MaAsLin 3 to discover DA and prevalence associations while accounting for compositionality, experimental protocols determining absolute abundance and new covariate types (including metatranscriptomics). On simulated datasets, MaAsLin 3 outperformed current state-of-the-art DA methods, maintaining precision as well as or better than other methods, even when its modeling assumptions were violated, and achieving higher recall in most simulations than any method with similar precision. Additionally, when estimating absolute abundance coefficients from relative abundance data, MaAsLin 3 and ANCOM-BC2 produced coefficients that were the most accurate (Fig. 1c and Extended Data Fig. 2). However, only MaAsLin 3 includes the ability to natively handle absolute abundance protocols (both spike-in and total biomass quantification) that are required to determine these coefficients experimentally. When applied to the HMP2 IBDMDB population of participants with and without IBD, 77% of taxonomic associations with IBD were quantified as prevalence associations, and 44 associations were new to MaAsLin 3 compared to MaAsLin 2, 8 of which were not previously significant due to opposing abundance and prevalence associations (Fig. 4).

MaAsLin 3's most prominent methodological improvements include median adjustment for compositional relative abundances and separate models for prevalence (presence/absence) and abundance associations. The former provides a simple way to improve on MaAsLin 2's minimal log transform that is equivalent to experimental absolute abundance data given reasonable assumptions, thus providing clear benefits. The latter can be beneficial both statistically and biologically. Statistically, modeling prevalence can help avoid the pitfalls of zero imputation for sparsity[26–28]. Furthermore, the predominance of prevalence associations in real data suggests that many taxa have abundances that are low enough or variable enough that only associations involving complete loss of the taxon or a reduction below the detection threshold are strong enough to be observed. Biologically, microorganisms with unique niches can exert phenotypic effects when simply present even at very low relative abundances, such as methanogenesis unique to the archaea[57] or the ability of *Clostridiodes difficile* to cause infections even in very low doses[16]. The presence of atypical species, regardless of abundance, can also identify instances of shared microbial communities either intentionally (for example, microbiota transplants) or unintentionally (for example, person-to-person transmission or contamination)[18]. Further, prevalence associations typically agreed with their corresponding abundance associations, so they might provide another measure of a taxon's general ability to survive in a particular environment.

Using MetaPhlAn 4 and MaAsLin 3, we identified a prevalence-only association between *D. welbionis* and IBD dysbiosis (Fig. 4c), underscoring the value of separating prevalence from abundance. *D. welbionis*, a recently isolated human commensal, has been linked to reduced weight gain in high-fat diet mouse models and is hypothesized to protect against metabolic disorders[58,59]. Given that creeping fat is a hallmark of some CD cases[60,61], and that *D. welbionis* is implicated in reducing adipose hypertrophy[58], its presence might mitigate this phenotype. It also metabolizes cholesterol via a protein homologous to IsmA, which converts cholesterol to coprostanol[62], a process diminished in IBD[63,64]. Thus, *D. welbionis* depletion may reduce cholesterol-to-coprostanol conversion and contribute to the elevated cardiovascular risk observed in IBD[65].

Despite MaAsLin 3's abilities to better detect abundance and prevalence associations and support new inference types, some limitations still remain. While MaAsLin 3 outperformed previous methods on essentially all simulations considered here, there remain other plausible microbial data distributions not yet tested. Additionally, many DA methods exist beyond those analyzed here; for brevity, we only compared against some of the most commonly used[9,10,12]. Notably, all DA methods remain affected by lower recall in small sample sizes and by the necessary limitations of finite read depths. Compared to MaAsLin 2 and ANCOM-BC2, MaAsLin 3 achieved lower recall on small sample sizes because logistic models often require more samples to reveal a significant association than linear models[66,67], and its linear modeling portion only uses the nonzero subset of the data (Extended Data Fig. 2). However, the slightly higher recall of these other methods in low sample sizes came at the cost of more biased coefficient estimates (MaAsLin 2) and severely reduced precision (ANCOM-BC2). Second, limited read depth prevented MaAsLin 3 from always distinguishing correctly between abundance and prevalence associations and sometimes caused it to miss associations. When a rare feature's abundance is associated with a covariate, that feature might be more likely to drop below the limit of detection depending on the covariate's value, yielding an entirely missed effect or a spurious prevalence effect in place of a true abundance effect. Rigorously accounting for this phenomenon would require experimental techniques such as culture-enriched molecular profiling[68] or highly tailored statistical techniques that, for example, distinguish the effects of read depth on the prevalence of taxa, genes, pathways and other features. In the absence of such methods, MaAsLin 3 reports prevalence associations deemed likely to be spurious and provides diagnostic plots for manual curation when necessary.

In summary, the methods introduced in MaAsLin 3 represent an important advancement in the accuracy, complexity and biological detail achievable in microbiome DA testing. They outperform state-of-the-art DA methods across a variety of simulations when testing for traditional associations, and MaAsLin 3 expands the space of possible associations with a variety of new covariate types. When applied to real adult and pediatric IBD datasets, MaAsLin 3 discovered hundreds of biologically plausible and internally consistent microbial associations. Many of these aligned with previous IBD findings, but some were only detectable with these updated models, which also clarified that most gut microbiome changes during inflammation corresponded with complete microbial loss. This informs, for example, strategies such as live biotherapeutic microbial supplementation in favor of, for example, prebiotic or dietary management in such settings[69]. MaAsLin 3 thus provides researchers with an important and much-improved set of capabilities for understanding microbiome associations with environmental phenotypes, human health and disease.

## Online content

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

## Methods

The MaAsLin 3 algorithm consists of (1) total-sum scaling feature abundance profiles to obtain relative abundances, (2) generating a prevalence (present versus absent) profile and retaining separately the nonzero abundances, (3) log transforming (base 2) the abundance dataset, (4) performing an augmented logistic regression on the prevalence dataset and linear regression on the abundance dataset, and (5) combining the logistic and linear effects into an overall effect for each feature–metadatum pair. This is in contrast to MaAsLin 2, which total-sum scales the feature abundance profiles, sets zeros to half the minimum observed abundance, log transforms all abundances (including imputed values) and performs a linear regression on the transformed values.

### MaAsLin 3 workflow

MaAsLin 3 requires, at a minimum, a samples-by-features table of feature abundances (counts or precomputed abundances for taxa, genes, and so on) and a samples-by-covariates table of metadata. The table of features is then normalized (by default, total-sum scaling to produce relative abundances), and features are optionally filtered by thresholds for prevalence, abundance and minimum variance for nonzero abundances. By default, any feature present with nonidentical abundance in at least two samples is retained. The abundance table is then split into a presence/absence table and a table of nonzero values that are subsequently transformed ($\log_2$ transformed by default). The specified logistic and linear models are fit on the presence/absence mask and transformed data, respectively, and the results are saved. A summary plot is produced for the top associations, and association-specific diagnostic plots are created for the top associations.

**Abundance modeling.** For the abundance associations, a linear model (possibly with mixed effects) is fit based on the specified formula one feature at a time. Following the notation of ANCOM-BC, let $i \in \{1, \dots, m\}$ be the feature index, $j \in \{1, \dots, g\}$ be the covariate index, and $k \in \{1, \dots, n\}$ be the sample index. Let $\hat{\beta}_{ij}^{\mathrm{rel}}$ be the slope corresponding to covariate $j$ when regressing the transformed relative abundance of feature $i$. When testing relative abundance associations with the median correction off, the $\hat{\beta}_{ij}$ are each tested against a null of zero according to standard homoskedastic regression theory. If the median comparison is on for testing absolute abundance associations (as by default), the median of the coefficients for each covariate $\hat{M}_j^{\mathrm{rel}} = \underset{i}{\mathrm{Median}}(\hat{\beta}_{ij}^{\mathrm{rel}})$ is calculated, and the coefficients $\hat{\beta}_{1j}^{\mathrm{rel}}, \dots, \hat{\beta}_{mj}^{\mathrm{rel}}$ are tested against $\hat{M}_j^{\mathrm{rel}}$ with a test that accounts for the variability in both the coefficients and the median (see 'Test implementation' in the Supplementary Information). Optionally, this median $\hat{M}_j^{\mathrm{rel}}$ is subtracted from each $\hat{\beta}_{ij}^{\mathrm{rel}}$.

In its normalization step, MaAsLin 3 can incorporate absolute abundance data from both spike-in experiments and total abundance estimation procedures. Let $A_{ik}$ be the absolute abundance of feature $i$ in sample $k$, let $T_k = \sum_i A_{ik}$ be the total absolute abundance in sample $k$, and let $P_{ik} = A_{ik}/T_k$ be the relative abundance of feature $i$ in sample $k$. Note that the ratios of relative abundances are equal to the ratios of absolute abundances[71]: $P_{ik}/P_{i'k} = A_{ik}/A_{i'k}$. In a spike-in experiment, estimators of both the absolute abundance $\hat{A}_{\mathrm{ref},k}$ and the relative abundance $\hat{P}_{\mathrm{ref},k}$ for the spike-in reference feature are known. Therefore, MaAsLin 3 can take as input the vector of absolute abundances of the spike-ins $[\hat{A}_{\mathrm{ref},1}, \dots, \hat{A}_{\mathrm{ref},n}]$ and use these to estimate the absolute abundances of the non-reference features as $\hat{A}_{ik} = \hat{A}_{\mathrm{ref},k} \cdot \hat{P}_{ik}/\hat{P}_{\mathrm{ref},k}$. Alternatively, MaAsLin 3 can take estimates $[\hat{T}_1, \dots, \hat{T}_n]$ of the samples' total abundances and estimate the absolute abundances as $\hat{A}_{ik} = \hat{P}_{ik} \cdot \hat{T}_k$. These absolute abundances are then filtered, $\log_2$ transformed, and regressed as in the relative abundance case but without the median comparison. As described previously[72], the spike-in procedure will eliminate bias due to sampling efficiency, while the total abundance scaling procedure will not.

Let $\hat{\beta}_{ij}^{\mathrm{abs}}$ be the slope corresponding to covariate $j$ when regressing the transformed absolute abundance of feature $i$. As described in 'With sparsity' in the Supplementary Information, when there is no sparsity, the absolute abundance slope can be decomposed as $\hat{\beta}_{ij}^{\mathrm{abs}} = \hat{\beta}_{ij}^{\mathrm{rel}} + \hat{\beta}_j^{\mathrm{tot}}$ where $\hat{\beta}_j^{\mathrm{tot}}$ is the slope corresponding to covariate $j$ in a regression of the $\log_2$-transformed per-sample total abundances on the same covariates. Since $\hat{\beta}_j^{\mathrm{tot}}$ does not depend on the feature $i$, the difference in two features' relative abundance coefficients is equal to the difference in the features' absolute abundance coefficients, and the orders of the slopes over the features are the same. Furthermore, as described in 'Relative and absolute coefficients' in the Supplementary Information, if at least half the features are not changing on the absolute scale ($\beta_{ij}^{\mathrm{abs}} = 0$), the test of $\beta_{ij}^{\mathrm{abs}} = 0$ for any individual coefficient is equivalent to a test of $\beta_{ij}^{\mathrm{rel}} = \underset{i}{\mathrm{Median}}(\beta_{ij}^{\mathrm{rel}})$, hence the test of the relative abundance coefficients against their median. Equivalent results hold even with unequal sampling efficiencies (see 'Extraction efficiency' in the Supplementary Information). These results have assumed no sparsity, but equivalent results can be obtained with sparsity if alternative, typically weaker, assumptions are satisfied (see 'With sparsity' in the Supplementary Information).

**Prevalence modeling.** For the prevalence associations, a logistic model, possibly with mixed effects, is fit based on the specified formula one feature at a time. For tractability, all zeros are treated as true feature absence, although such zeros could also arise from technical factors. By default, a data augmentation scheme is performed to avoid linear separability and outsized influence from a small number of data points. Let $\mathbf{B}_i$ be the $n \times 1$ binary vector of presence/absence values for feature $i$. Let $\mathbf{X}$ be the $n \times (g+1)$ fixed-effects design matrix from $g$ covariates and an intercept. The data augmentation procedure creates a $3n \times 1$ augmented presence/absence vector $\mathbf{B}_i^{\mathrm{aug}} = \begin{bmatrix} \mathbf{B}_i \\ \mathbf{1}_{n \times 1} \\ \mathbf{0}_{n \times 1} \end{bmatrix}$ and a $3n \times (g+1)$ augmented design matrix $\mathbf{X}^{\mathrm{aug}} = \begin{bmatrix} \mathbf{X} \\ \mathbf{X} \\ \mathbf{X} \end{bmatrix}$ and performs weighted logistic regressions with weights of 1 for the first $n$ (original) values and weights of $g/(2n)$ for the remaining (augmented) values. In this way, each original value is augmented with an extra presence and an extra absence such that the total weight of the augmented samples is equivalent to $g$ additional samples. This technique, equivalent to the Diaconis–Ylvisaker prior[36] has been proposed as an alternative to Firth regression with the advantage that it extends easily to mixed-effects models and model comparisons procedures.

When a low-abundance feature's abundance is associated with a metadatum, a spurious prevalence association can be induced by the abundance association if the feature regularly falls below the limit of detection despite being nonzero. The exact relationship between an abundance association and the induced prevalence association is complex, depending on both the (unknown) abundance association and the read depth. While a full treatment of this problem is beyond the scope of this paper, the following heuristic was implemented. After the prevalence associations are fit, a linear regression is performed on the log-transformed relative abundances, and the coefficients are tested against zero (not the median). Then, each prevalence association is flagged as likely abundance induced if (1) the corresponding abundance association has a significant $q$ value, (2) the sign of the abundance coefficient is the same as the sign of the prevalence coefficient, and (3) the magnitude of the abundance coefficient is larger than the magnitude of the prevalence coefficient.

Typically, read depth should be included as an untransformed covariate. Assuming the distribution of reads for a feature with relative abundance $r$ in a sample with $n$ reads is approximately $\mathrm{Binom}(n, r)$, the probability of the feature being sampled is $1 - (1 - r)^n$, so the log odds of inclusion are: $\log(\frac{1-(1-r)^n}{(1-r)^n})$. For $r$ small and $n$ large (for example, $n > 2/r$), $\log(\frac{1-(1-r)^n}{(1-r)^n}) \approx nr$, so the log odds of a feature being present are

approximately linear with the read depth, motivating the inclusion of the untransformed read depth covariate. When the read depth is independent of the other covariates (as in the simulations), it should not be included because of the non-collapsibility of the odds ratio. Otherwise, if it is associated with the other covariates as is often the case due to differences in sample biomass and extraction, it should be included.

**Combining associations.** To test for an overall association between a feature and a covariate, particularly for comparing against other tools that do not distinguish abundance from prevalence, an overall $P$ value is calculated as the Beta(1, 2) cumulative density function evaluated at the minimum of the two $P$ values from the abundance and prevalence models. Under the null of no abundance or prevalence association between the feature and the covariates, both models produce $P$ values uniform on 0 to 1 (ref. [73]). Furthermore, these $P$ values are independent as only the abundances in samples with the feature present are used to determine the abundance $P$ value, and knowing the feature is present (the only information used in the logistic $P$ value) gives no information about what its nonzero abundance is, under the null. Since the minimum of two independent Unif(0, 1) random variables has a Beta(1, 2) distribution, the minimum of the two $P$ values will have a Beta(1, 2) distribution under the null.

**New model components and specification capabilities.** MaAs-Lin 3 allows general mixed-effects models through the formula parsing of the package lme4 (ref. [70]), including interaction terms and random intercepts.

For omnibus testing of whether all coefficients corresponding to categories of a covariate are zero, an ANOVA-style test is performed. For linear models, this is an $F$-test (performed with the package lmerTest[74] for mixed-effects models), and for logistic models this is a likelihood-ratio test.

For testing differences in levels of an ordered covariate, contrast tests are performed between consecutive levels in the fit model with a right-hand side corresponding to zero or the median coefficient difference between the relevant levels. This test is performed with the package lmerTest for linear mixed-effects models and with the package multcomp[75] otherwise. No monotonic ordering is imposed in the model per se. The number of tests performed is equal to the number of levels minus one, and the $P$ values are FDR corrected with the $P$ values for all other coefficients.

Contrast tests are performed in the same way as the tests for ordered covariates but with arbitrary user-specified contrast matrices.

A feature-specific covariate table can be provided with dimensions matching the abundance table. Then, in the regression for feature $i$, the $i^{th}$ column of the table will be extracted and used as a covariate in the expanded $n \times g + 2$) design matrix **X**.

**Benchmarking on simulated microbial community data**
**SparseDOSSA 2.** The R package SparseDOSSA 2 was used to parametrically generate synthetic microbial community abundances according to templates informed by real data[5]. Using the 'Stool' template based on real microbiome data[76], SparseDOSSA 2 generated underlying null distributions of features from zero-inflated log-normal distributions. Then, synthetic metadata were created by sampling from a multivariate normal distribution and converting half the covariates to binary values. Using this synthetic metadata along with a synthetically generated table of associations to add in (effect sizes uniform on 2.5 to 5 unless otherwise specified), abundance (log-linear) and prevalence (logistic) associations were imposed on the absolute scale, and reads were sampled from the resulting relative abundances.

To evaluate MaAsLin 3's performance on group and ordered predictors, a multilevel categorical covariate was thermometer encoded in the synthetic metadata matrix (that is, for an ordered covariate with levels 1 through $P$ and an observed level of $j$, $P$ indicators are included with the first $j$ set to 1 and the rest set to 0). Next, coefficients were generated representing the total difference between the baseline level and the level with the greatest difference, and a Dirichlet (**1**) draw was used to divide the total difference among the levels. Then, with probability 1/2, each level's coefficient was set to 0 and added to the subsequent coefficient. Finally, the resulting metadata and coefficients were used to generate synthetic abundances, and the thermometer columns were collapsed into a single-ordered categorical covariate.

Natively, SparseDOSSA 2 does not simulate microbial feature spike-ins. Therefore, SparseDOSSA 2 was modified to include an extra feature in the absolute abundance generation step with an abundance chosen uniformly from 1% to 10% of the total abundance. The absolute abundance of this spiked-in feature was then stored and used in MaAs-Lin 3's spike-in mode. After this feature was added on the absolute scale, read sampling was performed as before.

**ANCOM-BC generator.** The data generation strategy from ANCOM-BC's benchmarking was also applied with minor modifications[10]. Using a template from a previously sequenced soil community as the null distribution[77], two groups of samples were created with the mean absolute abundances in one group multiplied by effect sizes drawn uniformly between 2.5 to 5 to create an unbalanced microbial load. Then, structural zeros were added in by setting a random 20% of features to 0 for all samples in a group. Finally, reads were sampled from the resulting profiles using rarefaction subsampling. This sampling used mock sequencing depths with the same distribution as in the ANCOM-BC evaluation but scaled to a mean depth of 50,000 reads for consistency with the other simulations. These evaluations are somewhat pathological insofar as a naive logistic regression fit on the prevalence data would be affected by linear separability, and the microbial loads of the two conditions are intentionally very unbalanced.

**Running DA tools on simulated datasets.** For simulations in which MaAsLin 3 was run on only relative abundance data, MaAsLin 3 (version 3.0.12) was almost always run with all default settings (median comparison on, data augmentation on, warning on abundance induced prevalence, no spike-ins) and a formula incorporating all simulated covariates (that is, a correctly specified model). The first exception was that for repeated sampling scenarios, per-participant random intercepts were also used. The second exception was that, in all relative abundance simulations, median subtraction was turned on (that is, subtracting the coefficient medians after testing against them) to evaluate the bias of the resulting differences as estimators of the synthetic absolute-scale coefficients. (By default in the software, this is off so that the relative abundance coefficients are returned for ease of interpretation.) When run using absolute abundance spike-in information, MaAsLin 3 was run with the spike-in absolute abundance provided as a parameter and the median comparison off.

ALDEx2 (version 1.36.0) was run with a model incorporating all the covariates and either the scale uncertainty $\gamma$ set to the default 0.5 or the scale parameter set to the total feature load calculated from the spike-in (scale-informed). Since ALDEx2 does not provide a random-effects option, to control for correlated measurements in simulations involving repeated sampling, per-participant fixed effects were fit and then removed from subsequent analysis (not included in the FDR correction). $P$ values were FDR corrected with the Benjamini–Hochberg procedure[78].

ANCOM-BC2 (version 2.4.0) was run with a model incorporating all the covariates and default settings except for (1) an FDR level of 0.1 rather than 0.05, (2) a prevalence threshold set to 0 rather than 0.1, and (3) the Benjamini–Hochberg FDR procedure rather than the Holm procedure, all for consistency with the other tools. Because it produced many false positives, the structural zero option was left off except in the ANCOM-BC generator evaluation when the assumptions for this

option were satisfied. Only significant associations that passed ANCOM-BC2's pseudo-count sensitivity screen were counted as significant. For repeated sampling scenarios, per-participant random intercepts were included. Coefficients were converted from their default $\log_e$ scale to the $\log_2$ scale for consistency with other tools in the analysis.

MaAsLin 2 (version 1.16.0) was run with a model incorporating all the covariates and default settings except for (1) an FDR level of 0.1 rather than 0.25 and (2) abundance and prevalence thresholds set to 0, both for consistency with the other tools. For repeated sampling scenarios, per-participant random intercepts were used.

DESeq2 (version 1.44.0) was run with a model incorporating all the covariates and default parameters except type was set to poscounts in estimateSizeFactors to handle the high sparsity of microbiome data. The tool edgeR (version 4.2.2) was run with a model incorporating all the covariates and default functions except glmFit rather than glmQLFit since the latter produced segmentation faults. Because neither of these tools provides a random-effects option, to control for correlated measurements in simulations involving repeated sampling, per-participant fixed effects were fit and then removed from subsequent analysis. $P$ values were FDR corrected with the Benjamini–Hochberg procedure.

### Benchmarking on real data
**Randomization procedure.** The binary metadata labels of the 38 datasets[1] were randomly shuffled to create 100 mock datasets for each real dataset. MaAsLin 3 was then run on both the randomized datasets and the original (non-randomized) datasets.

**Real absolute abundance data.** Three datasets with inferred absolute abundance data were analyzed: an infant gut dataset that used a spike-in procedure[37], a mouse diet dataset that used 16S digital PCR to estimate total microbial load[38] and an IBD/PSC dataset that used flow cytometry to estimate total microbial load[39]. One measurement per sample was included. For all three of these studies, estimated per-sample, per-taxon absolute abundance tables were already available, so the datasets were analyzed without further normalization in MaAsLin 3 to obtain absolute abundance coefficients. That is, equivalent results could have been obtained by supplying the spike-in abundances or total microbial load estimates to MaAsLin 3 along with the relative abundance tables, but since the absolute abundance scaling operations MaAsLin 3 would have performed were already precomputed, the data were used directly. For the infant dataset, the model included days since birth and read depth as fixed effects and infant ID as a random effect. For the mouse diet dataset, the model included diet and day as fixed effects and mouse ID as a random effect (read depth was not included because all samples had equal read depths). ANCOM-BC2 produced errors when the mouse ID was included as a random effect, so it was removed from the model. (This removal seemed to have a minimal effect since ANCOM-BC2's coefficients still achieved essentially perfect consistency with MaAsLin 3 in this setting). For the IBD/PSC dataset, diagnosis, age, gender, body mass index and read depth were included as fixed effects with diagnosis as a categorical variable that compared PSC-only, PSC-UC, CD-only and PSC-CD against healthy controls as the baseline. Read depth was included as a covariate in the real datasets because deeper sequencing will often associate with higher prevalence (taxa are detected more often with more reads), and this deeper sequencing could be confounded with the key sample covariates.

**HMP2 IBDMDB analysis.** The HMP2 IBDMDB dataset derives from 130 individuals recruited in five US medical centers with CD, UC or no IBD, who donated stool samples for one year, up to 24 times each[23]. Raw sequencing data from the 1,637 metagenomic samples were cleaned of host contaminant reads and low-quality reads with KneadData[41] using default parameters. Next, MetaPhlAn 4.0.6 (Oct22

database)[42] was used to construct taxonomic profiles with species genome bins for increased coverage, and the MetaPhlAn 3.0 profiles were downloaded from IBDMDB[23]. The resulting abundances were analyzed with each tool with antibiotic usage, diagnosis and dysbiosis status as fixed effects and participant ID as a random effect. One measurement per sample was included. Dysbiosis status is intended to reflect IBD disease flares; a dysbiosis score was calculated as the median Bray–Curtis dissimilarity to non-IBD individuals (excluding samples that came from the same participant), and the 90th percentile of these dissimilarity scores was set as the cutoff to define samples that were dysbiotic[23]. When not already split by pediatric and adult populations, age was also included as a fixed effect. Based on previous benchmarking presented herein, associations were only considered significant if they had a $q$ value below 0.1, a coefficient with an absolute value greater than 1 and no model-fitting errors. For each feature–metadatum pair, the significant association was considered to overlap between the adult and pediatric populations if the association's more significant coefficient (abundance or prevalence) from each population had the same sign.

Using the subset of samples from participants with CD, abundances were regressed in MaAsLin 3 using a model that incorporated categorical dietary frequency information (dietary component eaten in the last 7 days, in the last 4–7 days, and so on) as a group or ordered predictor. Also included in this model were dysbiosis, antibiotic usage, age and read depth as fixed effects and participant ID as a random intercept.

Previously computed pathway abundances from metagenomic and metatranscriptomic data using HUMAnN 3 (ref. 41) were downloaded from IBDMDB[23]. The abundances were preprocessed in MaAsLin 3 by, for each pathway in each sample, (1) using the relative abundances as-is if the DNA abundance was nonzero, (2) using the RNA abundance as-is and setting the DNA abundance to the $\log_2$ of the minimum nonzero DNA relative abundance in the dataset divided by 2 if the RNA was nonzero but the DNA was zero, or (3) excluding the observation if both the DNA and RNA were zero. Case 1 matches the usual assumptions of MaAsLin 3; case 2 assumes that the pathway DNA abundance was below the limit of detection; and case 3 assumes the pathway was not present at all, so there is no expression information. Using MaAsLin 3, the metatranscriptomic relative abundances were regressed on age, antibiotic usage, diagnosis and dysbiosis status as fixed effects, participant ID as a random effect, and pathway DNA relative abundance as a covariate-specific fixed effect, following previous recommendations[40]. Only relative abundance associations were of interest for the metatranscriptomics analysis, so no median comparison was performed.

### Reporting summary
Further information on research design is available in the Nature Portfolio Reporting Summary linked to this article.

### Data availability
The 38 real 16S rRNA gene sequencing datasets and their accompanying metadata used in the randomization procedure[1] are available at https://doi.org/10.6084/m9.figshare.14531724.v1 (ref. 79). The metadata and bioBakery 3 outputs for the IBDMDB data are available at https://ibdmdb.org/downloads/html/products_MGX_2017-08-12.html (merged table tab, taxonomic_profiles_3.tsv.gz and pathabundances_3.tsv.gz files). All other data including the synthetic datasets, the IBDMDB MetaPhlAn 4 profiles and the three datasets with inferred absolute abundance data[37–39] are available at https://figshare.com/s/8c09e0f276b427f07af6 (ref. 80).

### Code availability
MaAsLin 3 is available as an R package through Bioconductor and at https://github.com/biobakery/MaAsLin3 and https://doi.org/10.5281/zenodo.16950584 (ref. 81). All scripts used in benchmarking and real

data analysis are available at https://github.com/willnickols/MaAs-Lin3_benchmark/ and https://doi.org/10.5281/zenodo.16950573 (ref. 81). Tutorials for MaAsLin 3 are available at https://github.com/biobakery/biobakery/wiki/MaAsLin3/ and https://github.com/biobakery/biobakery/wiki/MTX-model-3/. The software and versions used in the analysis are: R (4.3.0), MaAsLin 3 (3.0.12), ALDEx2 (1.36.0), ANCOM-BC2 (2.4.0), MaAsLin 2 (1.16.0), DESeq2 (1.44.0), edgeR (4.2.2) and SparseDOSSA (0.99.2).

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

## Acknowledgements

The computations in this paper were run in part on the FASRC Cannon cluster supported by the FAS Division of Science Research Computing Group at Harvard University. W.A.N. was supported on a training grant from the National Institute of General Medical Sciences (T32GM135117). The work was supported by the National Institute of Diabetes and Digestive and Kidney Diseases of the National Institutes of Health (R24DK110499) to C.H. and the National Institute of Allergy and Infectious Diseases (U19AI110820) to D. Rasko (to C.H.) and the Crohn's and Colitis Foundation Early Career grant (to K.N.T).

## Author contributions

C.H., J.T.N. and K.N.T. conceptualized the study. W.N. led the analysis and implementation. J.T.N., T.K. and S.M. aided in code implementation. J.S. contributed to visualization. J.T.N. aided in code review. J.T.N., K.N.T., C.H., H.M. and E.A.F. contributed to results interpretation.

## Competing interests

C.H. is on the scientific advisory boards for Seres Therapeutics, Empress Therapeutics and ZOE Nutrition. The other authors declare no competing interests.

## Additional information

**Extended data** is available for this paper at https://doi.org/10.1038/s41592-025-02923-9.

**Correspondence and requests for materials** should be addressed to Curtis Huttenhower.

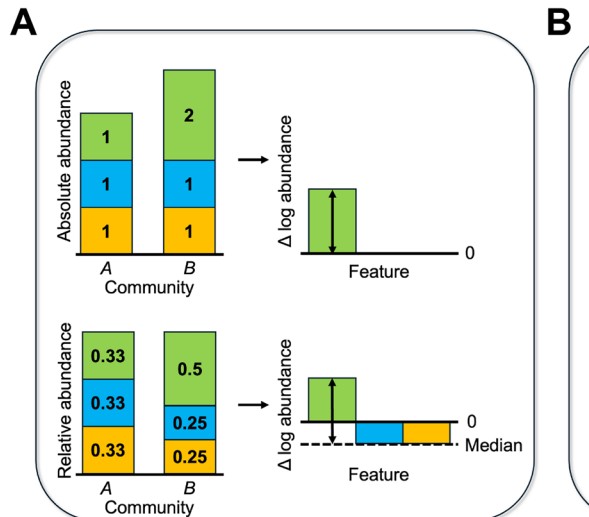

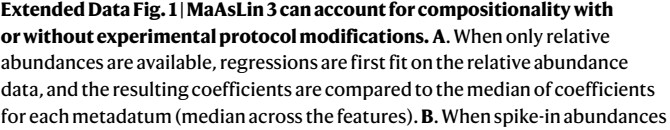

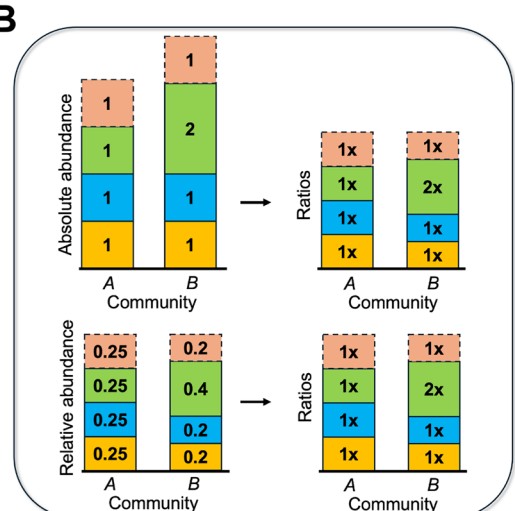

**Extended Data Fig. 1 | MaAsLin 3 can account for compositionality with or without experimental protocol modifications. A**. When only relative abundances are available, regressions are first fit on the relative abundance data, and the resulting coefficients are compared to the median of coefficients for each metadatum (median across the features). **B**. When spike-in abundances are known from the experimental protocol, the relative abundances are scaled to the spike-in to compute ratios. These ratios are then used in the regressions. Alternatively, the relative abundances can be scaled by a measure of total community abundance, producing estimated absolute abundances that are then used in the regressions.

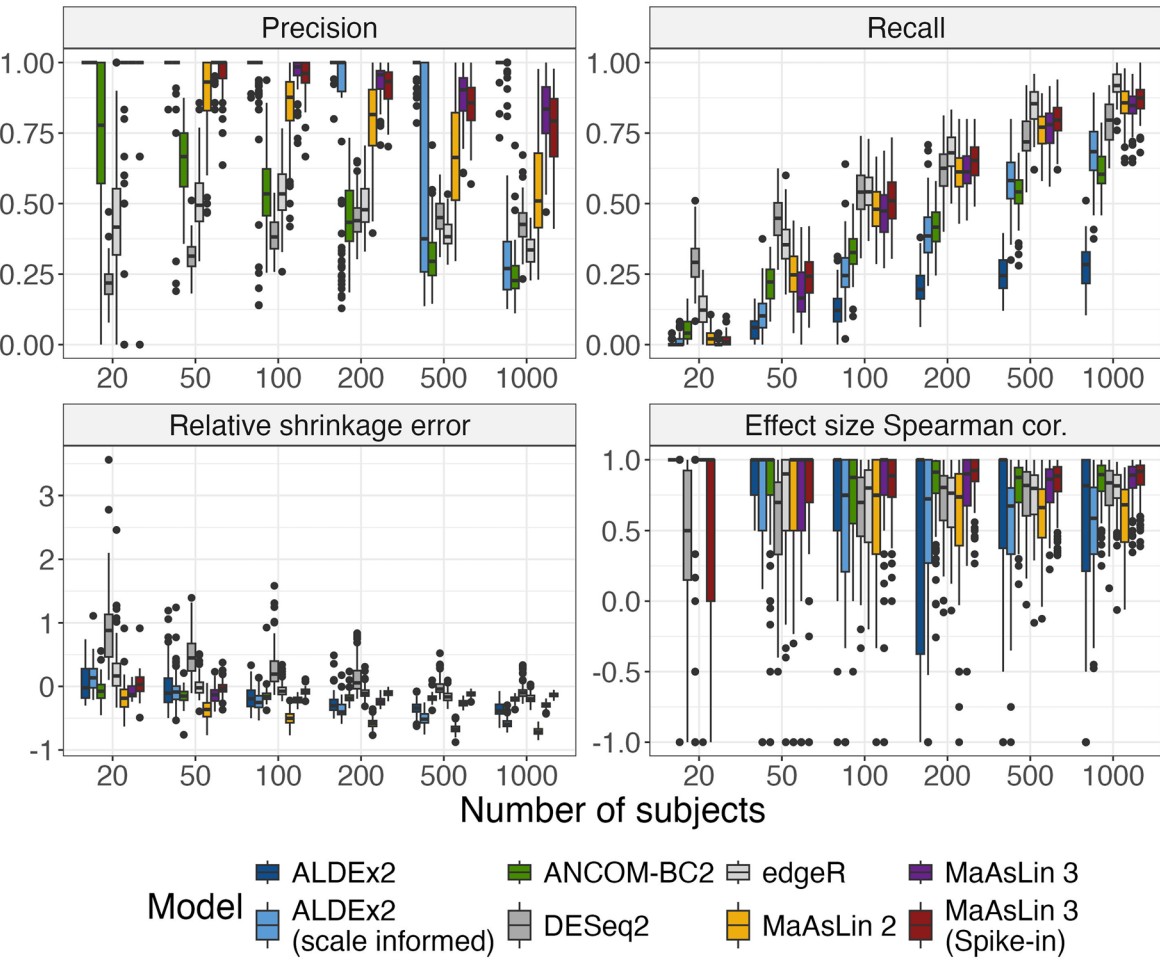

**Extended Data Fig. 2 | MaAsLin 3 improves accuracy over other DA methods, particularly with high sample sizes.** MaAsLin 3 and other common DA methods were run on the 100 synthetic log-normal datasets 2 from Fig. 1c. Each metric was calculated as before. 1 is optimal for all metrics except shrinkage, for which 0 is optimal. Each point represents a simulated dataset. Boxplots display the median, interquartile range, and whiskers extending to the most extreme values within 1.5 × IQR and individual points indicating outliers.

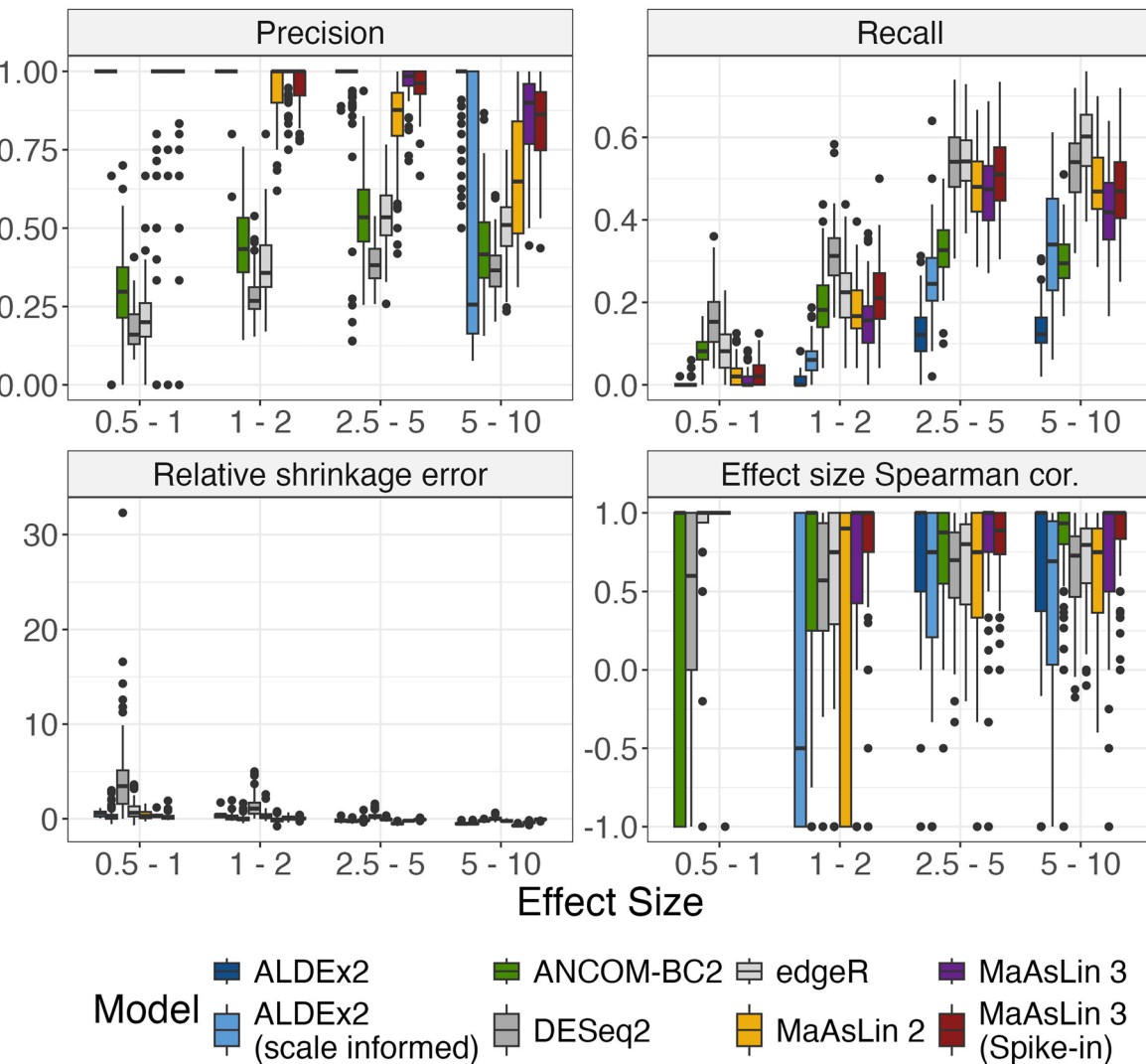

**Extended Data Fig. 3 | MaAsLin 3 maintains high precision and accurate effect size estimation over a range of biologically relevant effect sizes.** MaAsLin 3 and other common DA methods were run on 100 synthetic log-normal datasets from SparseDOSSA 2. The datasets were generated as in Fig. 1c but with 100 samples for all datasets and varying effect sizes. Each metric was calculated as before. 1 is optimal for all metrics except shrinkage, for which 0 is optimal. Each point represents a simulated dataset. Boxplots display the median, interquartile range, and whiskers extending to the most extreme values within 1.5x IQR and individual points indicating outliers.

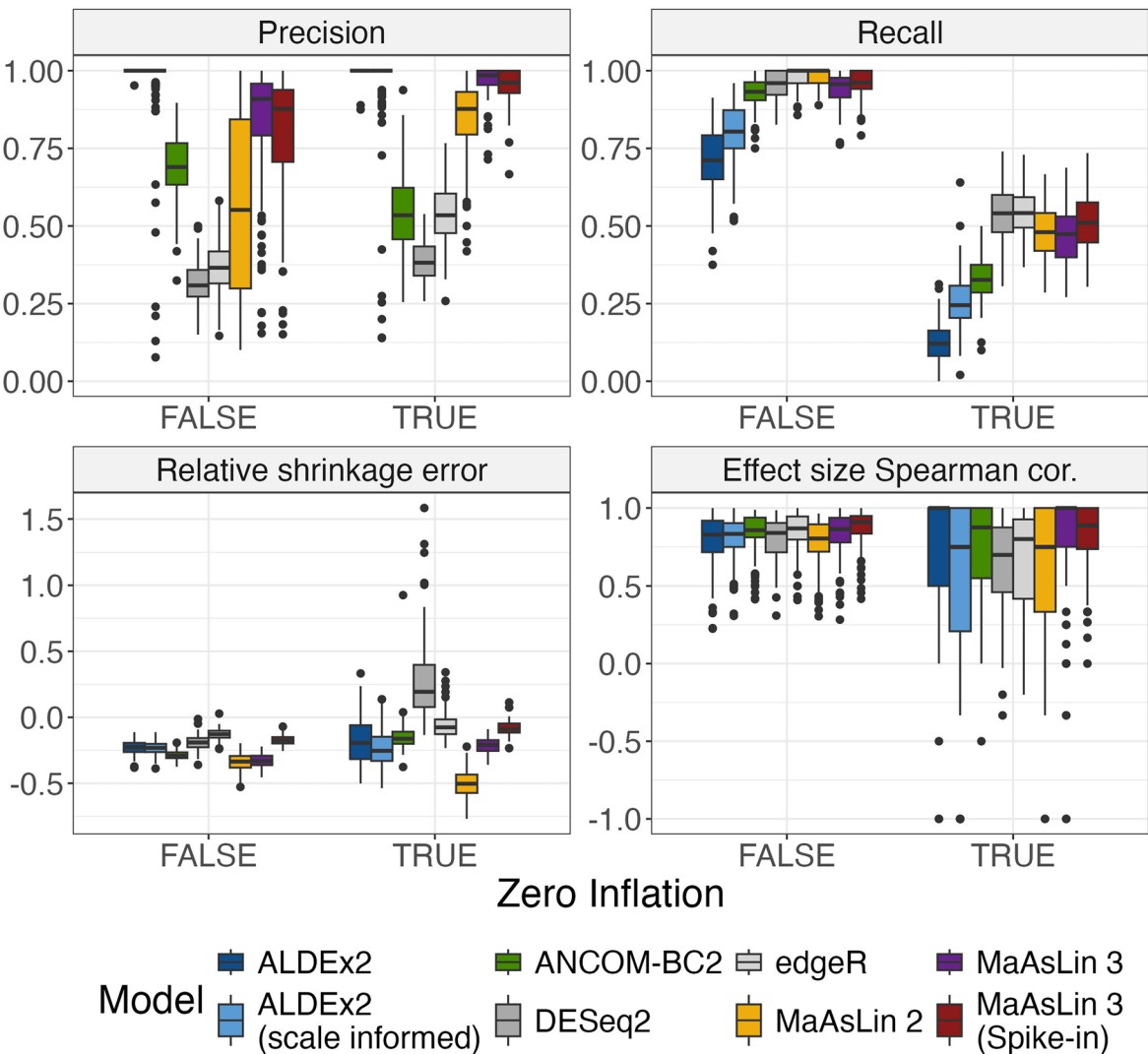

**Extended Data Fig. 4 | MaAsLin 3 maintains high precision regardless of whether zeros are due to sequencing or true absence.** MaAsLin 3 and other common DA methods were run on 100 synthetic log-normal datasets from SparseDOSSA 2. The datasets were generated as in Fig. 1c but with 100 samples for all datasets and either no zero inflation (sequencing zeros only) or zero inflation (both sequencing and biological zeros). Each metric was calculated as before. 1 is optimal for all metrics except shrinkage, for which 0 is optimal. Each point represents a simulated dataset. Boxplots display the median, interquartile range, and whiskers extending to the most extreme values within 1.5x IQR and individual points indicating outliers.

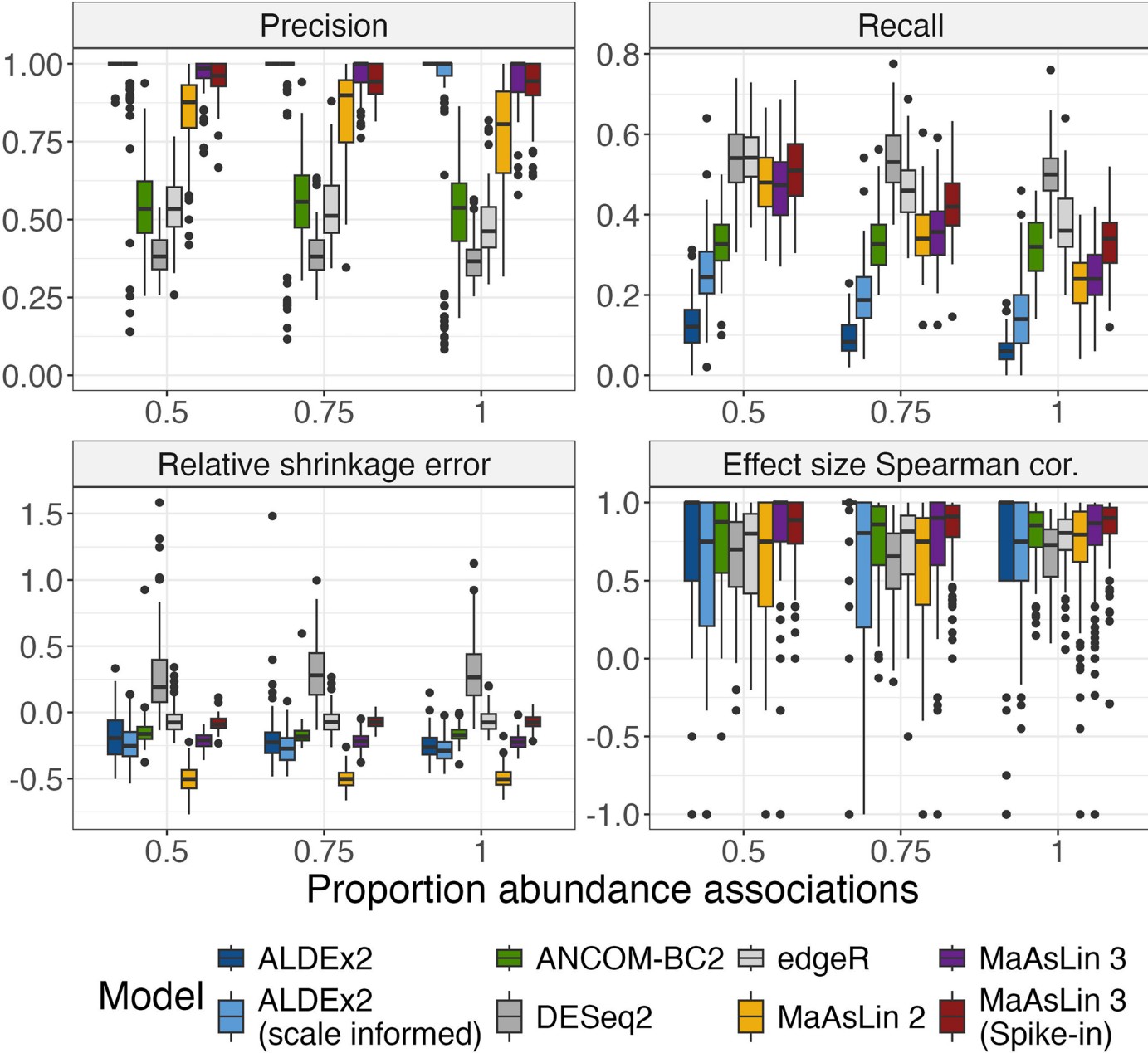

**Extended Data Fig. 5 | MaAsLin 3 maintains high precision regardless of the proportion of associations are with abundance rather than prevalence.** MaAsLin 3 and other common DA methods were run on 100 synthetic lognormal datasets from SparseDOSSA 2. The datasets were generated as in Fig. 1c but with 100 samples for all datasets and a varying proportion of associations with abundance rather than prevalence. Each metric was calculated as before. 1 is optimal for all metrics except shrinkage, for which 0 is optimal. Each point represents a simulated dataset. Boxplots display the median, interquartile range, and whiskers extending to the most extreme values within 1.5x IQR and individual points indicating outliers.

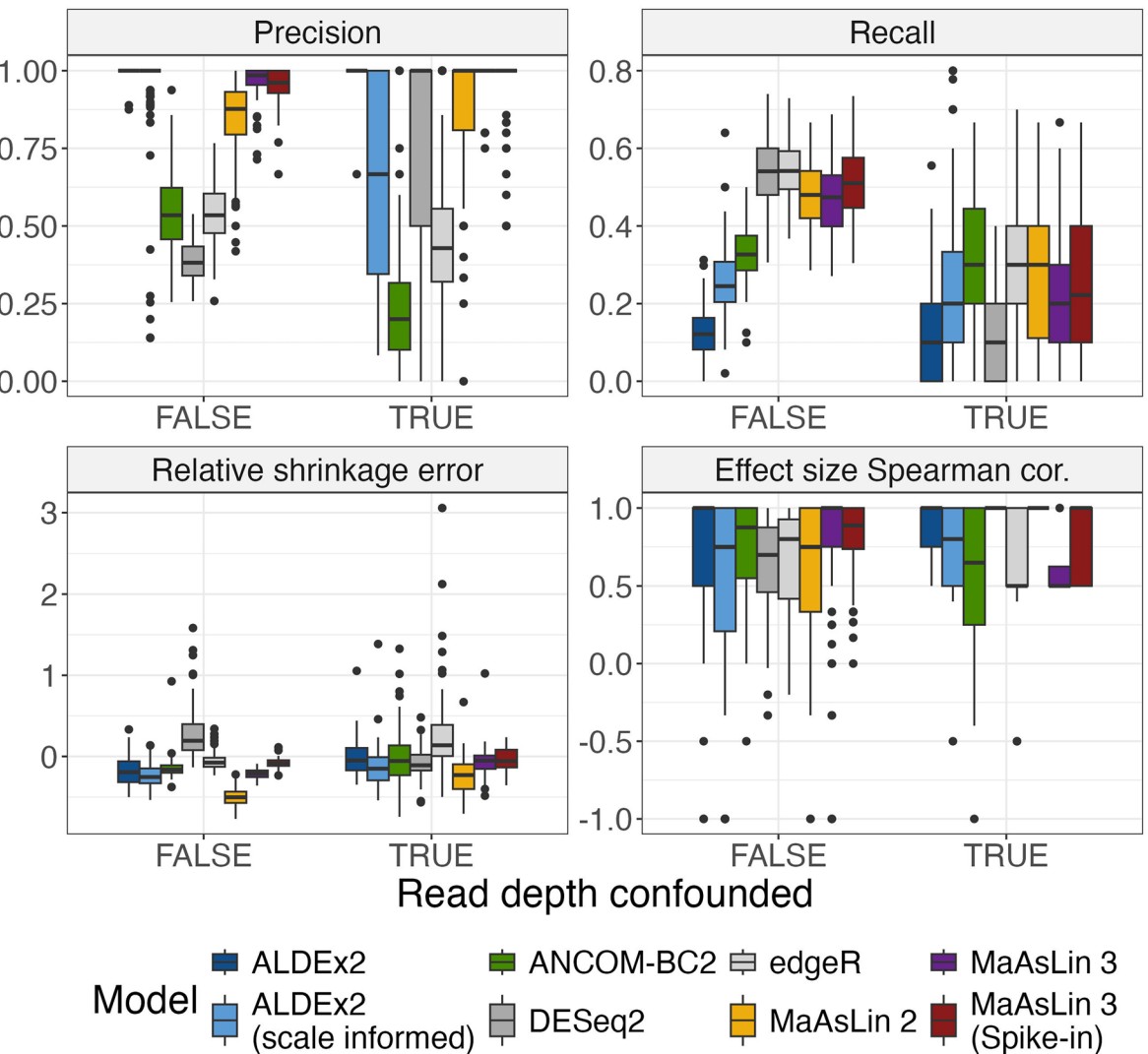

**Extended Data Fig. 6 | Including read depth as a covariate prevents spurious associations from metadata correlated with read depth.** MaAsLin 3 and other common DA methods were run on 100 synthetic log-normal datasets from SparseDOSSA 2. The datasets were generated as in Fig. 1c but with 100 samples for all datasets and either the typical five metadata uncorrelated with read depth or one metadatum correlated with read depth. In the correlated case, read depth was included as a covariate in all models. Each metric was calculated as before. 1 is optimal for all metrics except shrinkage, for which 0 is optimal. Each point represents a simulated dataset. Boxplots display the median, interquartile range, and whiskers extending to the most extreme values within 1.5x IQR and individual points indicating outliers.

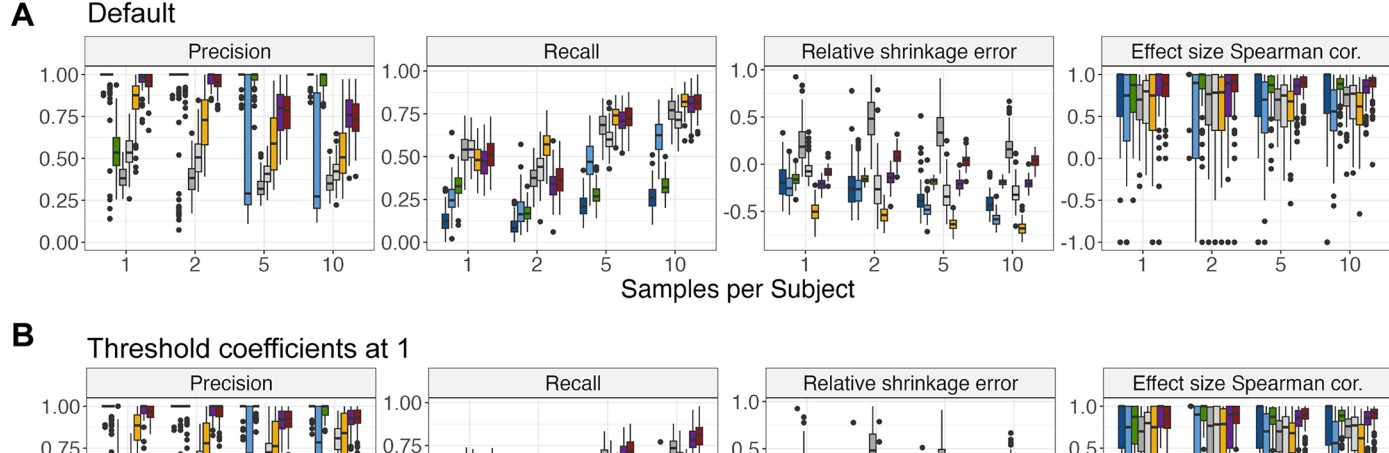

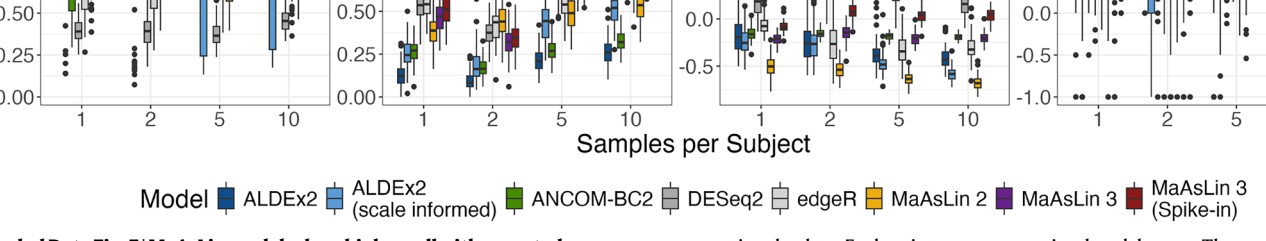

**Extended Data Fig. 7 | MaAsLin models show high recall with repeated sampling at the cost of reduced precision, though precision can be substantially improved for MaAsLin 3 by thresholding fit coefficients.** MaAsLin 3 and other common DA methods were run on 100 synthetic log-normal datasets from SparseDOSSA 2. Each dataset was generated as in Fig. 1c but with 100 subjects and varying numbers of samples per subject. For each feature, each subject was given a random intercept drawn from a normal distribution when generating the data. Each point represents a simulated dataset. The metrics were calculated as before on either all associations (**A**) or only associations with fit coefficients larger than 1 in absolute value (**B**). 1 is optimal for all metrics except shrinkage, for which 0 is optimal. Each point represents a simulated dataset. Boxplots display the median, interquartile range, and whiskers extending to the most extreme values within 1.5x IQR and individual points indicating outliers.

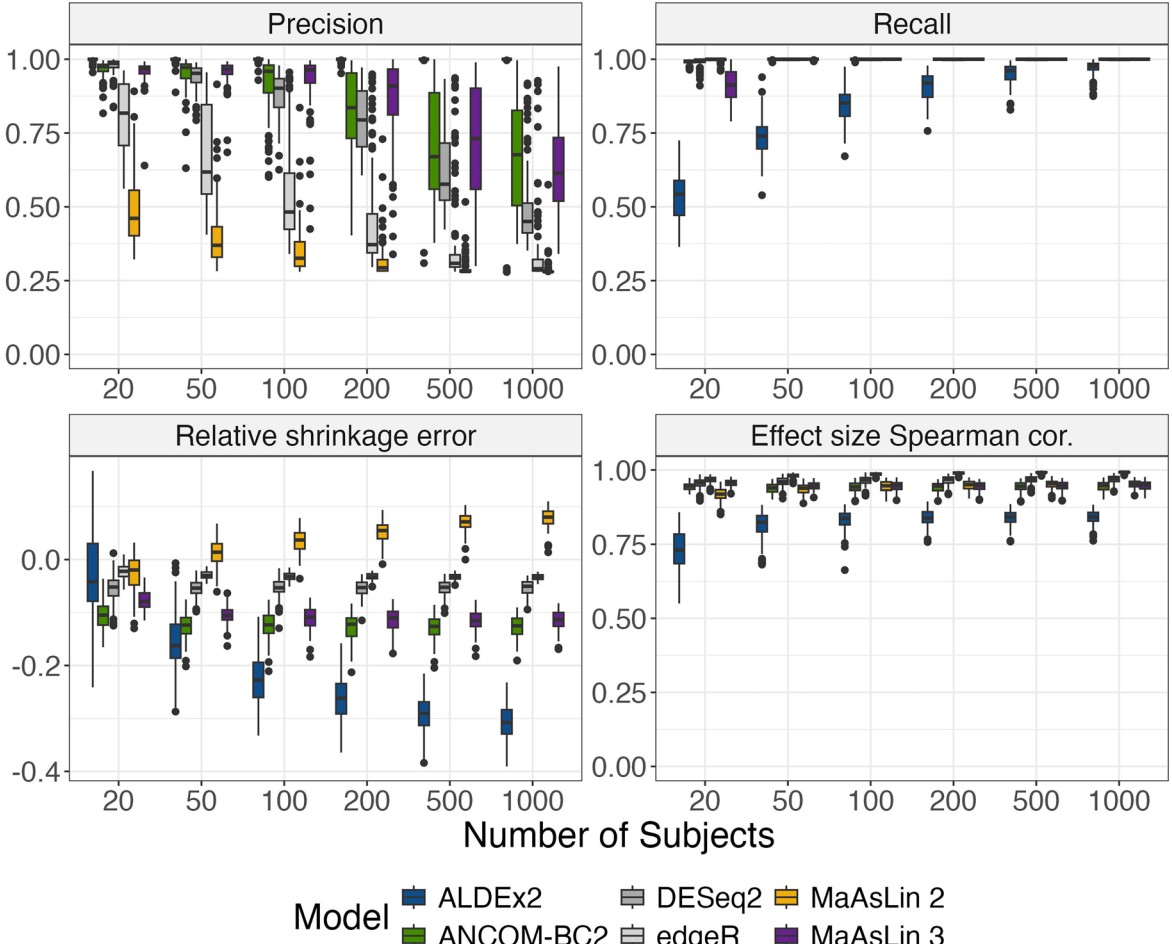

**Extended Data Fig. 8 | MaAsLin 3 maintains or improves accuracy even when its modeling assumptions are violated.** MaAsLin 3 and other common DA methods were run on 100 synthetic datasets generated with the 'soil' option of the ANCOM-BC evaluation. For these simulations, 1000 features and 2 groups were simulated with 10% of the feature-metadatum pairs having true associations with coefficients uniform from 2.5 to 5, half of which were positive and half of which were negative. Additionally, 20% of the features were set to have structural zeros in which all samples from one group lacked the feature. Highly skewed and unbalanced read depths (analogous to 16S read count) were drawn using the ANCOM-BC evaluation procedure with a mean depth of 50,000. Metrics were computed as before. 1 is optimal for all metrics except shrinkage, for which 0 is optimal. Each point represents a simulated dataset. Boxplots display the median, interquartile range, and whiskers extending to the most extreme values within 1.5x IQR and individual points indicating outliers.

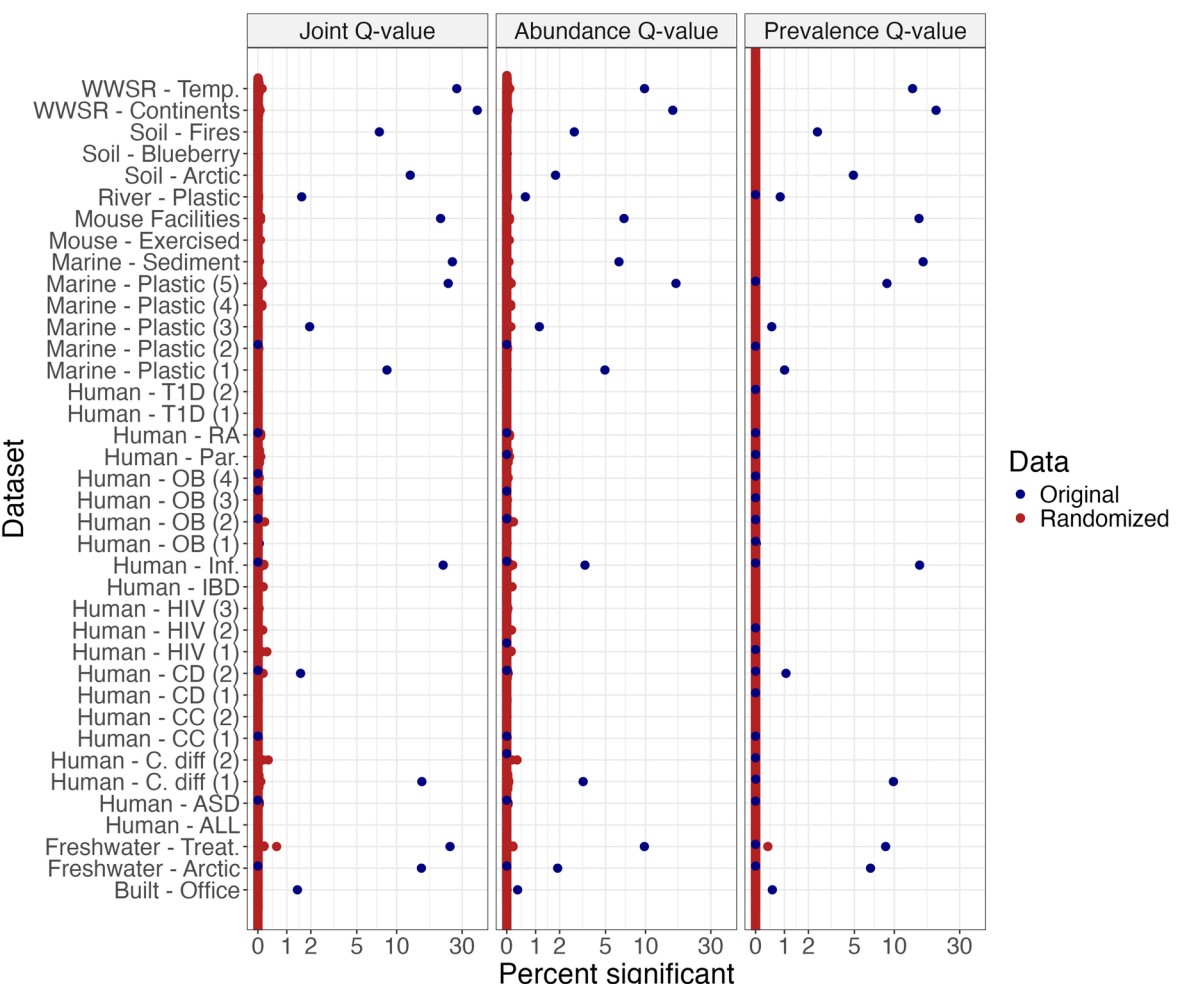

**Extended Data Fig. 9 | A randomization test using real data shows that MaAsLin 3 almost never produces false positives when all associations are null.** Using previous datasets, 100 mock datasets were created for each real dataset by permuting the binary metadata labels. MaAsLin 3 was then run on the resulting randomized metadata and ASV tables, which should have no associations. For comparison, MaAsLin 3 was also run on the original datasets without randomization, which should have associations if they exist. The percent of significant feature-metadatum associations for both schemes is displayed. 0 is optimal for all randomized settings.

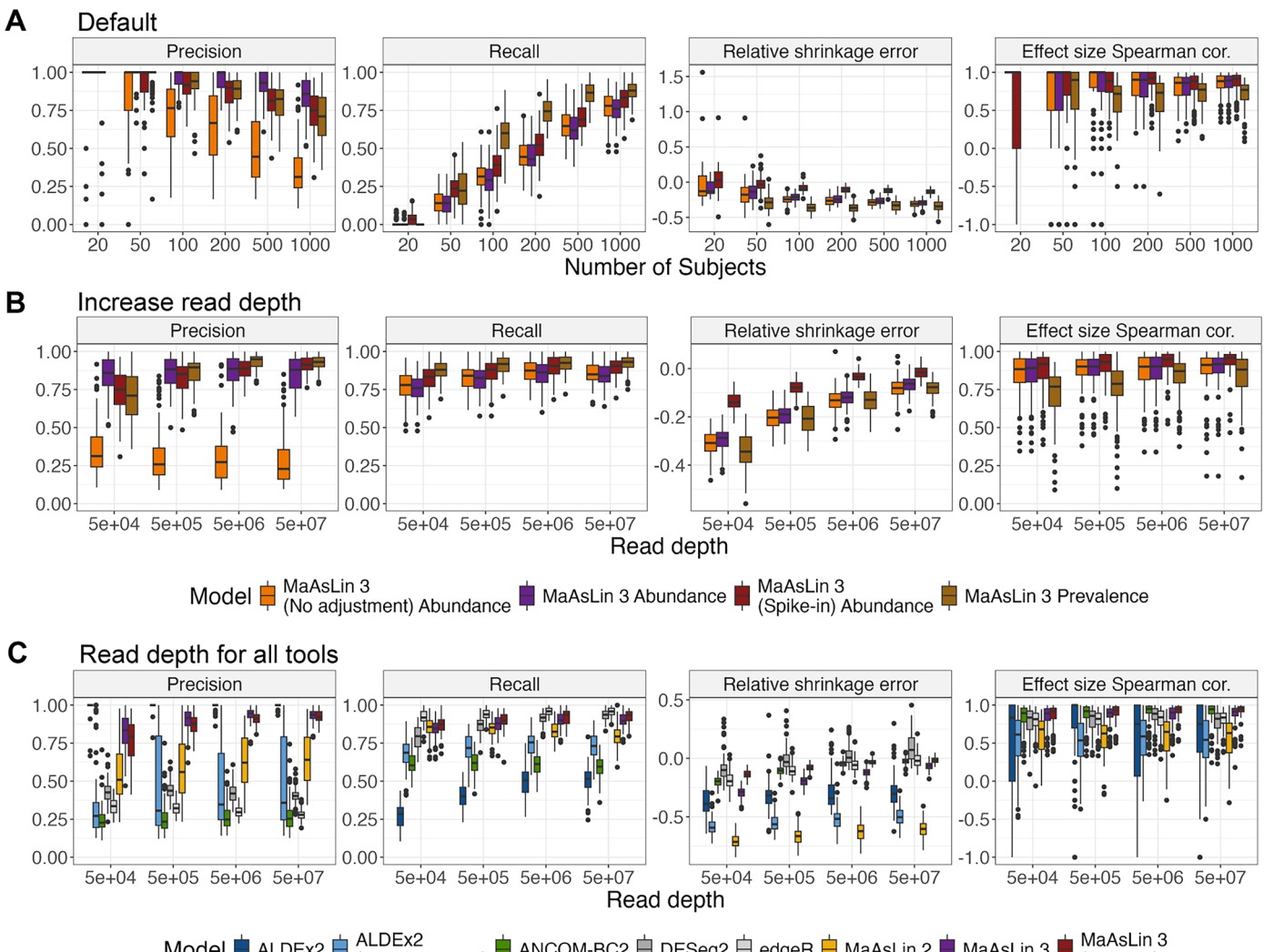

**Extended Data Fig. 10 | Precision loss with high power can be mitigated by increasing read depth. A**. Using 100 synthetic log-normal datasets from SparseDOSSA 2, MaAsLin 3 was run with no median adjustment for compositionality (abundance only), with the default adjustment (abundance only), with synthetic spike-in information (abundance only), and with the default prevalence setting. For A, the same datasets were used as for Fig. 1c. Significant associations (individual q-value less than 0.1) were only considered correct if they matched the true associations in the feature, metadatum, and type of association (prevalence/abundance). The relative shrinkage error and effect size correlation were computed as before. 1 is optimal for all metrics except shrinkage, for which 0 is optimal. Each point represents a simulated dataset.

**B**. Datasets were generated as in A but holding the number of subjects fixed at 1000 and varying the mean read depth. **C**. The same metrics were evaluated for all methods on the datasets from B. In this evaluation, significant associations (q-value less than 0.1, joint q-value for MaAsLin 3) were considered correct if they matched the true associations in the feature and metadatum. A mismatch in association type—abundance versus prevalence—was allowed for all methods since no methods besides MaAsLin 3 report association type. Each point represents a simulated dataset. Boxplots display the median, interquartile range, and whiskers extending to the most extreme values within 1.5x IQR and individual points indicating outliers.

# Reporting Summary

## Statistics

For all statistical analyses, confirm that the following items are present in the figure legend, table legend, main text, or Methods section.

| n/a | Confirmed | |
|---|---|---|
| ☐ | ☒ | The exact sample size (*n*) for each experimental group/condition, given as a discrete number and unit of measurement |
| ☐ | ☒ | A statement on whether measurements were taken from distinct samples or whether the same sample was measured repeatedly |
| ☐ | ☒ | The statistical test(s) used AND whether they are one- or two-sided<br>*Only common tests should be described solely by name; describe more complex techniques in the Methods section.* |
| ☐ | ☒ | A description of all covariates tested |
| ☐ | ☒ | A description of any assumptions or corrections, such as tests of normality and adjustment for multiple comparisons |
| ☐ | ☒ | A full description of the statistical parameters including central tendency (e.g. means) or other basic estimates (e.g. regression coefficient) AND variation (e.g. standard deviation) or associated estimates of uncertainty (e.g. confidence intervals) |
| ☐ | ☒ | For null hypothesis testing, the test statistic (e.g. *F*, *t*, *r*) with confidence intervals, effect sizes, degrees of freedom and *P* value noted<br>*Give P values as exact values whenever suitable.* |
| ☒ | ☐ | For Bayesian analysis, information on the choice of priors and Markov chain Monte Carlo settings |
| ☒ | ☐ | For hierarchical and complex designs, identification of the appropriate level for tests and full reporting of outcomes |
| ☐ | ☒ | Estimates of effect sizes (e.g. Cohen's *d*, Pearson's *r*), indicating how they were calculated |

*Our web collection on statistics for biologists contains articles on many of the points above.*

## Software and code

Policy information about availability of computer code

| Data collection | Datasets were manually downloaded from their source locations (see Data) with no code. |
|---|---|
| Data analysis | All code used to analyze data is provided at https://github.com/WillNickols/maaslin3_benchmark/. The versions of software used in the analysis are: R (4.3.0), MaAsLin 3 (3.0.12), ALDEx2 (1.36.0), ANCOM-BC2 (2.4.0), MaAsLin 2 (1.16.0), DESeq2 (1.44.0), edgeR (4.2.2), SparseDOSSA (0.99.2). |

For manuscripts utilizing custom algorithms or software that are central to the research but not yet described in published literature, software must be made available to editors and reviewers. We strongly encourage code deposition in a community repository (e.g. GitHub). See the Nature Portfolio guidelines for submitting code & software for further information.

## Data

Policy information about availability of data

All manuscripts must include a data availability statement. This statement should provide the following information, where applicable:

- Accession codes, unique identifiers, or web links for publicly available datasets
- A description of any restrictions on data availability
- For clinical datasets or third party data, please ensure that the statement adheres to our policy

The 38 real datasets used in the randomization procedure are available at https://figshare.com/articles/dataset/16S_rRNA_Microbiome_Datasets/14531724. The metadata and bioBakery 3 outputs for the IBDMDB data are available at https://ibdmdb.org/downloads/html/products_MGX_2017-08-12.html (Merged Table tab,

taxonomic_profiles_3.tsv.gz and pathabundances_3.tsv.gz files). All other data including the synthetic datasets, the IBDMDB MetaPhlAn 4 profiles, and the three datasets with inferred absolute abundance data are available at https://figshare.com/s/8c09e0f276b427f07a.

# Research involving human participants, their data, or biological material

Policy information about studies with human participants or human data. See also policy information about sex, gender (identity/presentation), and sexual orientation and race, ethnicity and racism.

| | |
|---|---|
| Reporting on sex and gender | N/A |
| Reporting on race, ethnicity, or other socially relevant groupings | N/A |
| Population characteristics | N/A |
| Recruitment | N/A |
| Ethics oversight | N/A |

Note that full information on the approval of the study protocol must also be provided in the manuscript.

# Field-specific reporting

Please select the one below that is the best fit for your research. If you are not sure, read the appropriate sections before making your selection.

☒ Life sciences        ☐ Behavioural & social sciences        ☐ Ecological, evolutionary & environmental sciences

For a reference copy of the document with all sections, see nature.com/documents/nr-reporting-summary-flat.pdf

# Life sciences study design

All studies must disclose on these points even when the disclosure is negative.

| | |
|---|---|
| Sample size | Infant gut dataset: 178 infants, 650 samples<br>Mouse diet dataset: 12 mice, 45 samples<br>IBD/PSC dataset: 170 participants, 170 samples<br>IBDMDB: 130 participants, 1637 samples<br><br>The inclusion of 4 datasets with 2502 samples was deemed sufficient because the included datasets capture a diverse range of topics relevant to human health, and the number and size of included datasets is similar to or greater than that of other similar methods development and evaluation publications. |
| Data exclusions | In the IBD/PSC dataset, samples with missingness in diagnosis, age, gender, BMI, FC, or CRP were excluded to avoid introducing complexities when comparing across differential abundance tools while simultaneously dealing with missing data and because proper clinical effects were not the intention of this section. Likewise, participants with a UC diagnosis only (4 after earlier filtering) were excluded because there were too few of these participants to make accurate comparisons between differential abundance tools. |
| Replication | Reproducible workflows for the evaluations are available at https://github.com/WillNickols/maaslin3_benchmark/. The analysis of 38 binary datasets was performed as in Nearing et al. 2022, and the findings were similar (independent replication of previous results). The IBDMDB analyses were performed with both MaAsLin 3 and MaAsLin 2 since MaAsLin 2 had been applied to these data beforehand, and the results were largely consistent (independent replication of previous results). For all other analyses, simulations were performed 100 times each and seeds were set for exact reproducibility, but no independent replication was performed. |
| Randomization | No randomization was performed since no causal claims were addressed. For the infant dataset, the linear model included days since birth and read depth as fixed effects and infant ID as a random effect. For the mouse diet dataset, the model included diet and day as fixed effects and mouse ID as a random effect (read depth was not included because all samples had equal read depths). For the IBD/PSC dataset, diagnosis, age, gender, BMI, and read depth were included as fixed effects with diagnosis as a categorical variable that compared PSC-only, PSC-UC, CD-only, and PSC-CD against healthy controls as the baseline. Read depth was included as a covariate in the real datasets because deeper sequencing will often associate with higher prevalence (taxa are detected more often with more reads), and this deeper sequencing could be confounded with the key sample covariates.<br><br>The IBDMDB dataset was analyzed with a linear model incorporating antibiotic usage, diagnosis, and dysbiosis status as fixed effects and participant ID as a random effect. When not already split by pediatric and adult populations, age was also included as a fixed effect. Using the subset of samples from participants with CD, abundances were regressed using a model that incorporated categorical dietary frequency information as a group or ordered predictor. Also included in this model were dysbiosis, antibiotic usage, age, and read depth as fixed effects and participant ID as a random intercept. Metatranscriptomic relative abundances were regressed on age, antibiotic usage, diagnosis, and dysbiosis status as fixed effects, participant ID as a random effect, and pathway DNA relative abundance as a covariate-specific fixed effect. |
| Blinding | Blinding was not relevant since all datasets were previously published. |

# Reporting for specific materials, systems and methods

We require information from authors about some types of materials, experimental systems and methods used in many studies. Here, indicate whether each material, system or method listed is relevant to your study. If you are not sure if a list item applies to your research, read the appropriate section before selecting a response.

## Materials & experimental systems

| n/a | Involved in the study |
|-----|----------------------|
| ☒ ☐ | Antibodies |
| ☒ ☐ | Eukaryotic cell lines |
| ☒ ☐ | Palaeontology and archaeology |
| ☒ ☐ | Animals and other organisms |
| ☒ ☐ | Clinical data |
| ☒ ☐ | Dual use research of concern |
| ☒ ☐ | Plants |

## Methods

| n/a | Involved in the study |
|-----|----------------------|
| ☒ ☐ | ChIP-seq |
| ☒ ☐ | Flow cytometry |
| ☒ ☐ | MRI-based neuroimaging |

## Plants

**Seed stocks**

*Report on the source of all seed stocks or other plant material used. If applicable, state the seed stock centre and catalogue number. If plant specimens were collected from the field, describe the collection location, date and sampling procedures.*

**Novel plant genotypes**

*Describe the methods by which all novel plant genotypes were produced. This includes those generated by transgenic approaches, gene editing, chemical/radiation-based mutagenesis and hybridization. For transgenic lines, describe the transformation method, the number of independent lines analyzed and the generation upon which experiments were performed. For gene-edited lines, describe the editor used, the endogenous sequence targeted for editing, the targeting guide RNA sequence (if applicable) and how the editor was applied.*

**Authentication**

*Describe any authentication procedures for each seed stock used or novel genotype generated. Describe any experiments used to assess the effect of a mutation and, where applicable, how potential secondary effects (e.g. second site T-DNA insertions, mosiacism, off-target gene editing) were examined.*

