## [Peer Review File · Nature Methods]

MaAsLin 3: Refining and extending generalized multivariable linear models for meta-omic association discovery

Corresponding Author: Dr Curtis Huttenhower

Version 0:

Decision Letter:

5th Feb 2025

Dear Dr. Huttenhower,

Your Article, "MaAsLin 3: Refining and extending generalized multivariable linear models for meta-omic association discovery", has now been seen by 2 reviewers. As you will see from their comments below, although the reviewers find your work of potential interest, they have raised a number of concerns. We are interested in the possibility of publishing your paper in Nature Methods, but would like to consider your response to these concerns before we reach a final decision on publication.

We therefore invite you to revise your manuscript to fully address all these concerns.

Link Redacted

We hope to receive your revised paper within 3 months. If you cannot send it within this time, please let us know. In this event, we will still be happy to reconsider your paper at a later date so long as nothing similar has been accepted for publication at Nature Methods or published elsewhere.

OPEN SCIENCE REQUIREMENTS

REPORTING SUMMARY AND EDITORIAL POLICY CHECKLISTS

EXTENDED DATA FIGURES

DATA AVAILABILITY

All novel DNA and RNA sequencing data, protein sequences, genetic polymorphisms, linked genotype and phenotype data, gene expression data, macromolecular structures, and proteomics data must be deposited in a publicly accessible database, and accession codes and associated hyperlinks must be provided in the "Data Availability" section.

CODE AVAILABILITY

Please include a "Code Availability" subsection in the Online Methods which details how your custom code is made available. Only in rare cases (where code is not central to the main conclusions of the paper) is the statement "available upon request" allowed (and reasons should be specified).

For more information on our code sharing policy and requirements, please see: <https://www.nature.com/nature-research/editorial-policies/reporting-standards#availability-of-computer-code>

MATERIALS AVAILABILITY

More details about our materials availability policy can be found at <https://www.nature.com/nature-portfolio/editorial->

ORCID

Sincerely,

Lin Tang, PhD
Senior Editor
Nature Methods

Reviewers' Comments:

Reviewer #1 (Remarks to the Author):

Nickols et al. propose an updated version of MaAslin3 that integrates prevalence modeling, abundance modeling, and absolute abundance modeling when spike-in data is available. This approach leverages disjoint pieces of information to enhance precision in differential abundance modeling. Overall, the study rigorously benchmarks the proposed methodology through simulations and makes a compelling case for the advantages of incorporating prevalence modeling and spike-ins. Below, I provide detailed comments regarding benchmarks, comparisons to other methods, and choices related to model specification:

1. Simulations

Benchmarking on Real Data: While simulations are excellent for debugging and controlled comparisons, they often fail to capture the full complexity of biological data. Is it feasible to benchmark these differential abundance methods using real datasets? For instance, approaches such as discriminative balances proposed by Quinn et al. (PMC7141889) or Pinto et al. (DOI: 10.1128/msystems.00053-18) may provide additional insights.

2. Methodology

Justification for Distribution Choices: Why weren't count-based distributions (e.g., negative binomial, multinomial) used? These distributions naturally account for read depth and could mitigate false positives, especially for low-abundance taxa. If read depth isn't properly modeled, it might confound results, leading to spurious findings.

Prevalence Filter vs. Read Depth: By using count-based distributions, reliance on the prevalence filter might be reduced. Additionally, pseudocount dependency becomes less critical, as log-fold changes can be regularized effectively without inflating to extreme values. Alternatively, incorporating read depth uncertainty directly (e.g., via normal approximations to binomial/multinomial distributions) could strengthen the model's robustness.

Benchmarks and Comparisons: A direct comparison to DESeq2 and edgeR, which are well-established differential abundance methods, would be valuable. Demonstrating performance differences in the context of read depth and pseudocount sensitivity would provide a clearer understanding of the advantages of this methodology.

3. Counterintuitive Results

The statement, "When applying a q-value threshold of 0.1, the default MaAsLin3 abundance model achieved higher precision than MaAsLin3 with spike-in normalization," seems counterintuitive. Spike-ins are generally expected to improve precision by reducing technical noise. Could this be due to suboptimal calibration of the spike-in normalization? Further investigation or clarification is warranted here.

4. ALDEx2 Benchmarking

When benchmarking against ALDEx2, were scale-variant priors employed? Using scale-variant priors could enable a more direct, apples-to-apples comparison with the spike-in normalization applied here. See the recent work (bioRxiv: 2023.10.21.563431v1) for reference.

5. Figure 4

The addition of a comparison against existing differential abundance methods would be informative. Specifically, showing that this method identifies biologically meaningful signals that other approaches fail to detect would highlight its utility and innovation.

Reviewer #1 (Remarks on code availability):

The code seems well documented.

Reviewer #2 (Remarks to the Author):

The analysis of differential abundance (DA) to identify microbial features associated with traits of interest is a fundamental question in microbiome research. In this study, the authors introduce a novel DA tool that applies a modified logistic regression to prevalence data and a linear regression to abundance data, claiming superior performance compared to existing methods. The methodology is clearly articulated, the simulation benchmarks are comprehensive, and the real-world application is well presented. However, I have several questions and suggestions that I hope will further refine the text and enhance the technical aspects of the study.

General comments

1. Handling of Zeros. Separating prevalence modeling from abundance modeling is one of the key innovations presented in this work. The authors aim to avoid the common pitfall of adding pseudo-counts to the data. However, my concern is that this approach implicitly assumes that data sparsity arises primarily from biological factors. In reality, technical factors—such as insufficient sequencing depth—could be a dominant contributor to the observed sparsity. This issue is reflected in the simulation results.

Using the authors' provided code, I modified the sparseDOSSA2 simulator to output the true absolute abundance (with and without spike-in). Under the simulation settings presented in this manuscript, I compared the proportion of zeros in the true absolute abundance versus the observed abundance. The results indicate that they are nearly identical—over 90% of the zeros in the observed data originate from absolute abundance sparsity rather than under-sampling. This suggests that the observed zeros are largely "true" biological zeros rather than artifacts of under-sampling.

Since this setting directly favors prevalence modeling, I wonder if an alternative simulation benchmark could be conducted to evaluate the method's performance when most zeros result from under-sampling. Additionally, I would appreciate the authors' perspectives on how to distinguish between different types of zeros—either within this study or as a direction for future research.

2. Effect Sizes of the Abundance Data. This comment builds upon my previous point regarding the handling of zeros. Based on the Methods section and the provided code, the true effect sizes are defined as the log fold changes of the "true" nonzero abundance. This approach may put competing methods at a disadvantage, as they do not explicitly differentiate between different types of zeros. This is particularly relevant for methods that incorporate pseudo-counts, where zeros are treated as technical zeros to some extent.

I wonder whether this distinction contributes to the poorer performance of competing methods—such as ALDEx2—in terms of relative shrinkage error and effect size correlation. It may also impact their precision and recall. Again, introducing an alternative benchmark that explicitly accounts for technical zeros could provide additional insights into the performance of different methods.

3. Assumption on the Median Absolute Abundance Coefficient. As the authors acknowledge, the assumption that at least half of the features remain unchanged with respect to a given metadatum may not always hold in certain scenarios. However, the current simulation benchmark primarily evaluates effect size bias and correlation. I believe that precision and recall could serve as more informative metrics in these extreme cases. It would be valuable to see if the authors could provide this additional evaluation to further assess the impact of this assumption.

Minor comments

1. Clarifications on Level-vs-Level Differences. Since the authors mention "ordered" levels, I wonder whether a trend test function is incorporated into the main method. If not, does the method rely solely on pairwise comparisons? Additionally, how do users address the more severe multiple comparisons issue in this context? Clarification on this aspect would be helpful.

Version 1:

Decision Letter:

Our ref: NMETH-A59105A

14th May 2025

Dear Dr. Huttenhower,

Thank you for submitting your revised manuscript "MaAsLin 3: Refining and extending generalized multivariable linear models for meta-omic association discovery" (NMETH-A59105A). It has now been seen by the original referees and their comments are below. The reviewers find that the paper has improved in revision, and therefore we'll be happy in principle to publish it in Nature Methods, pending minor revisions to satisfy the referees' final requests and to comply with our editorial and formatting guidelines.

Please supply a point-by-point response letter when submitting the revised version of the paper.

We are now performing detailed checks on your paper and will send you a checklist detailing our editorial and formatting

requirements within two weeks or so. Please do not upload the final materials and make any revisions until you receive this additional information from us.

TRANSPARENT PEER REVIEW

Nature Methods offers a transparent peer review option for new original research manuscripts submitted from 17th February 2021. We encourage increased transparency in peer review by publishing the reviewer comments, author rebuttal letters and editorial decision letters if the authors agree. Such peer review material is made available as a supplementary peer review file. **Please state in the cover letter ‘I wish to participate in transparent peer review’ if you want to opt in, or ‘I do not wish to participate in transparent peer review’ if you don’t.** Failure to state your preference will result in delays in accepting your manuscript for publication.

Please note: we allow redactions to authors’ rebuttal and reviewer comments in the interest of confidentiality. If you are concerned about the release of confidential data, please let us know specifically what information you would like to have removed. Please note that we cannot incorporate redactions for any other reasons. Reviewer names will be published in the peer review files if the reviewer signed the comments to authors, or if reviewers explicitly agree to release their name. For more information, please refer to our [FAQ page](https://www.nature.com/documents/nr-transparent-peer-review.pdf).

ORCID

Sincerely,

Lin Tang, PhD
Senior Editor
Nature Methods

Reviewer #1 (Remarks to the Author):

My comments have been addressed

Reviewer #1 (Remarks on code availability):

no comments

Reviewer #1 (Remarks on figshare data availability):

no comments

Reviewer #2 (Remarks to the Author):

I thank the authors for carefully addressing my previous comments. The manuscript has improved considerably and is now more comprehensive.

1. I appreciate the authors’ in-depth explanation regarding zeros arising from both technical and biological factors. This response is informative and, in my view, would strengthen the manuscript and benefit readers if included in the Discussion section. However, as the authors rightly note, the assumption that “data sparsity arises primarily from biological factors” is often unverifiable. It would be appropriate to either acknowledge this as a limitation in the Discussion or clearly state it as a key modeling assumption in the Methods section.

2. I am not sure if I fully agree with the statement, in the rebuttal letter, that “ANCOM-BC2 does not rely on pseudo-counts.” According to the user manual, after classifying zero types, differential abundance analysis is performed on taxa not deemed to contain structural zeros—i.e., those whose sparsity is attributed to technical factors such as under-sampling. Pseudo-counts are then added to these technical zeros during a sensitivity analysis. This implies that, once a taxon passes structural zero screening, the presence or absence of zeros still influences significance determination. This deviates from MaAsLin3’s assumption that “only the abundances in samples with the feature present are used to determine the abundance p-value.” This distinction may help explain why, in Subfigure 4, ANCOM-BC2 showed improved precision (since most zeros are now technical) but still yielded some false positives. I recommend clarifying this in the Discussion or Methods section to better guide users on the underlying assumptions of these methods.

3. As a final major point, I encourage the authors to consider including a false discovery rate benchmark using real datasets. One approach would be to generate null data by permuting covariates or applying a similar randomization procedure, so that all findings would, by construction, represent false discoveries. The authors have already implemented this for MaAsLin3; extending it to include other methods would provide a more empirical and assumption-light benchmark for comparing FDR control.

Minor Comments:

1. The statement that “ANCOM-BC simulations are somewhat pathological with respect to both sparsity and compositionality” is rather strong. It would be helpful if the authors could elaborate on this characterization or clarify what specifically makes the simulations “pathological.”

2. Optional suggestion: Some studies (e.g., <https://link.springer.com/article/10.1186/s40168-024-01822-z>) suggest that the earlier version of ANCOM-BC, or ANCOM-BC2 applied to data with pre-added pseudo-counts (e.g., +1), may achieve better FDR control under certain scenarios. If deemed useful, the authors might consider including ANCOM-BC or ANCOM-BC2 with pre-added pseudo-counts as an additional comparator in the benchmarking analysis.

Version 2:

Decision Letter:

21st Oct 2025

Dear Dr Huttenhower,

I am pleased to inform you that your Article, "MaAsLin 3: Refining and extending generalized multivariable linear models for meta-omic association discovery", has now been accepted for publication in Nature Methods. The received and accepted dates will be 13th Dec 2024 and 21st Oct 2025. This note is intended to let you know what to expect from us over the next month or so, and to let you know where to address any further questions.

Over the next few weeks, your paper will be copyedited to ensure that it conforms to Nature Methods style. Once your paper is typeset, you will receive an email with a link to choose the appropriate publishing options for your paper and our Author Services team will be in touch regarding any additional information that may be required. It is extremely important that you let us know now whether you will be difficult to contact over the next month. If this is the case, we ask that you send us the contact information (email, phone and fax) of someone who will be able to check the proofs and deal with any last-minute problems.

Authors may need to take specific actions to achieve compliance with funder and institutional open access mandates.

If your research is supported by a funder that requires immediate open access (e.g. according to [a href="https://www.springernature.com/gp/open-science/plan-s-compliance"> Plan S principles](https://www.springernature.com/gp/open-science/plan-s-compliance) or the [a href="https://www.springernature.com/gp/open-science/us-federal-agency-compliance"> NIH public access policy](https://www.springernature.com/gp/open-science/us-federal-agency-compliance)) then you should select the gold OA route, and we will direct you to the compliant route where possible. Because authors warrant under our subscription licensing terms that they haven't committed to licensing any version of their article under a licence inconsistent with the terms of our agreement – including the applicable embargo period – publication under the subscription model isn't suitable for authors whose funders require no embargo.

You can now use a single sign-on for all your accounts, view the status of all your manuscript submissions and reviews,

access usage statistics for your published articles and download a record of your refereeing activity for the Nature journals.

Please feel free to contact me if you have questions about any of these points. Thank you very much for publishing your paper with Nature Methods!

Best regards,

Lin Tang, PhD
Senior Editor
Nature Methods

** Visit the Springer Nature Editorial and Publishing website at http://editorial-jobs.springernature.com?utm_source=ejP_NMeth_email&utm_medium=ejP_NMeth_email&utm_campaign=ejp_Nmeth for more information about our career opportunities. If you have any questions please click [here](mailto:editorial.publishing.jobs@springernature.com).

Open Access This Peer Review File is licensed under a Creative Commons Attribution 4.0 International License, which permits use, sharing, adaptation, distribution and reproduction in any medium or format, as long as you give appropriate credit to the original author(s) and the source, provide a link to the Creative Commons license, and indicate if changes were made. In cases where reviewers are anonymous, credit should be given to 'Anonymous Referee' and the source.

Dear Editor,

We are very grateful to you and to both referees for the thoughtful feedback on this manuscript, and for your suggestions for improving it during review. We have carefully considered all comments and revised the manuscript with appropriate changes. The most significant of these include:

- We have expanded the set of simulations to evaluate MaAsLin 3, including:
 - A simulation comparing the performance of differential abundance tools with (original) and without (new) zero inflation due to biological rather than sequencing zeros. While the zero inflated model is more consistent with real data, MaAsLin 3 performed well in both scenarios and maintained higher precision than any other tool with similar recall.
 - A simulation in which an increasing proportion of the associations were due to abundance, not prevalence, differences.
 - A simulation in which the metadatum of interest is confounded with read depth, in which MaAsLin's inclusion of read depth as a covariate prevents false positives due to the confounding.
- We have updated the tools against which we compare MaAsLin 3, which maintained its favorable results:
 - ALDEx2 is now run both with and without scale models, either using no external information (i.e., only the default scale uncertainty) or incorporating scale information through the spike-ins as MaAsLin 3 would.
 - At the reviewers' request, we have also included the RNA differential abundance tools edgeR and DESeq2, which have historically been used in this context.
 - Several comparisons now also include precision and recall metrics.
- We clarify for Reviewer 1 that MaAsLin 3 does not incorporate pseudocounts into its model.
- In addition to these revisions based on review feedback, we have also made several substantive updates to the software itself based on feedback from early users of the tool, among other ease-of-use updates:
 - Added a parameter that allows users to control for small groups with random intercepts in the abundance model and with fixed intercepts in the prevalence model. This better handles models with random effects and small groups for logistic regression.
 - Users can now control for species-level abundances rather than DNA gene-level abundances in metatranscriptomics experiments, consistent with methods that were previously found to perform well.

Detailed responses to each point raised by the reviewers are included here, along with corresponding excerpts from the revised manuscript. We attach two versions of the revised manuscript, one with modified sections highlighted and one without markup. We appreciate the feedback and opportunity to make these changes, and we are of course open to any further suggestions or discussion to finalize the manuscript.

Sincerely,
Curtis Huttenhower

Professor, Departments of Biostatistics and Immunology and Infectious Diseases
Harvard Chan Microbiome in Public Health Center
Harvard T.H. Chan School of Public Health
Associate Member, Broad Institute of MIT and Harvard

Reviewer 1

Nickols et al. propose an updated version of MaAslin3 that integrates prevalence modeling, abundance modeling, and absolute abundance modeling when spike-in data is available. This approach leverages disjoint pieces of information to enhance precision in differential abundance modeling. Overall, the study rigorously benchmarks the proposed methodology through simulations and makes a compelling case for the advantages of incorporating prevalence modeling and spike-ins. Below, I provide detailed comments regarding benchmarks, comparisons to other methods, and choices related to model specification:

Comments

R1.1.1 *Simulations. Benchmarking on Real Data: While simulations are excellent for debugging and controlled comparisons, they often fail to capture the full complexity of biological data. Is it feasible to benchmark these differential abundance methods using real datasets? For instance, approaches such as discriminative balances proposed by Quinn et al. (PMC7141889) or Pinto et al. (DOI: 10.1128/msystems.00053-18) may provide additional insights.*

¹Many thanks to the reviewer for the time taken to provide these comments on the manuscript, which have been very helpful during revision. While there are relatively few settings in which real-world data include a ground truth for method evaluation, our initial submission did evaluate all differential abundance tools on three diverse real datasets (from infants, mice, and adults) with experimentally derived absolute abundances, along with the Inflammatory Bowel Disease Multi'omics Database dataset consisting of 1,637 longitudinal stool samples. Furthermore, we applied MaAsLin 3 to 38 previously published two-group real datasets from a recent benchmarking study². MaAsLin 3's strong performance across these diverse datasets suggests that it is robust to many types of biological complexity, even if any particular simulation fails to capture that full complexity.

Simulations also, of course, provide a setting in which controlled comparisons can be made between methods, although we agree that each simulation strategy makes its own assumptions that might not accurately reflect true biological effects. Therefore, we evaluated with simulation strategies incorporating multiple different generation schemes including SparseDOSSA 2 and ANCOM-BC's evaluation method^{3,4}. SparseDOSSA in particular was designed to simulate microbial community dynamics by using real data to parameterize templates which can then be used to generate new synthetic communities while also allowing controlled changes in abundance and prevalence.

Finally, the two papers referenced by the reviewer seek to identify sets of taxa that are consistent with respect to each other across environments and use these sets to discriminate phenotypically labeled data. While regression coefficients can be used to select taxa that discriminate between phenotypes, this is not their primary purpose, and they are not optimized for this task. Therefore,

consistent with previous benchmarking studies, we focused our real data evaluations on distinguishing which features differed according to sample covariates and assessing the consistency of these results across tools.

R1.2.1 *Methodology. Justification for Distribution Choices: Why weren't count-based distributions (e.g., negative binomial, multinomial) used? These distributions naturally account for read depth and could mitigate false positives, especially for low-abundance taxa. If read depth isn't properly modeled, it might confound results, leading to spurious findings.*

In the paper describing MaAsLin 2, the predecessor to MaAsLin 3, log-normal methods were typically found to outperform count-based distributions in balancing specificity and sensitivity, leading us to focus on these methods during further development⁵. Additionally, count-based distributions do not accommodate non-integer data or those which are already relative abundances (e.g., from shotgun metagenomic sequencing), but log-normal methods can handle either by first converting the counts to relative abundances. Theoretically, log-normal models also benefit from properties of the Central Limit Theorem since the sampling distribution of the coefficients will be asymptotically normal regardless of the precise distribution of the residuals¹ (i.e., the approximation is better with more data). By contrast, because the negative binomial distribution imposes a mean-variance dependence, if the true data are not distributed as a negative binomial, more samples will lead to greater statistical power to reject null hypotheses related to the model, even if the rejection only signals model misspecification. Since the very high sparsity of microbiome data means that most count-based distributions will need zero-inflation components as well, which raise the same set of issues⁶, in practice this means that count-based methods can be extremely fragile and thus underperform⁷.

In order to account for read depth in this setting, we recommend including read depth as a covariate to ensure that prevalence associated with read depth is not misattributed to other biologically relevant covariates. To further address the concern that read depth might confound the results, we added a new comparison in which read depth was either correlated or not correlated with a metadatum of interest. We then compared the performance of the tested DA methods in either scenario, with the addition of read depth as a covariate when confounding occurred (**Sup. Fig. 6**). MaAsLin 3 maintained excellent precision in either case, though the recall of all tools decreased with confounding (as expected when true covariation cannot be associated with the “biological” covariate). We summarize these findings in the main text and **Sup. Fig. 6**:

When read depth was correlated with a metadatum of interest, including read depth as a covariate prevented precision loss, though all methods exhibited decreased recall with this correction (as expected, **Sup. Fig. 6**).

Supplementary Figure 6: Including read depth as a covariate prevents spurious associations from metadata correlated with read depth. MaAsLin 3 and other common DA methods were run on 100 synthetic log-normal datasets from SparseDOSSA 2. The datasets were generated as in Fig. 1C but with 100 samples for all datasets and either the typical five metadata uncorrelated with read depth or one metadata correlated with read depth. In the correlated case, read depth was included as a covariate in all models. Each metric was calculated as before. 1 is optimal for all metrics except shrinkage, for which 0 is optimal. Each point represents a simulated dataset.

R1.2.2 *Prevalence Filter vs. Read Depth: By using count-based distributions, reliance on the prevalence filter might be reduced. Additionally, pseudocount dependency becomes less critical, as log-fold changes can be regularized effectively without inflating to extreme values. Alternatively, incorporating read depth uncertainty directly (e.g., via normal approximations to binomial/multinomial distributions) could strengthen the model's robustness.*

We apologize for any confusion regarding pseudocounts and would like to clarify that, unlike other differential abundance methods, MaAsLin 3 does not use pseudocounts and is therefore not dependent on them to regularize log-fold changes. In our evaluations, the linear abundance

models were never in need of regularization (pseudo-counts or otherwise), and only the logistic regression was mildly regularized with data augmentation (not pseudo-counts) to avoid linear separability.

We would also like to clarify that read depth is never uncertain itself—it is known exactly based on the sequencing results—so the only uncertainty to address is uncertainty in whether a zero is due to absence of a taxon or low abundance of that taxon. Furthermore, a normal approximation to the binomial ($\text{Bin}(n = \text{reads}, p = \text{relative abundance})$) or multinomial would only be valid if np was large. However, for taxa in which stochastic drop-out is relevant, np is near zero (hence the propensity to drop-out), so a normal approximation would not be valid. As shown in **Sup. Fig. 6**, including read depth as a covariate is sufficient to ensure the model is robust to spurious associations between metadata and read depth that might otherwise manifest as false positives.

That being said, because sparsity is very high in metagenomic data (66%, 80%, and 97% zeros in the mouse, IBD/PSC, and infant datasets analyzed respectively) in contrast to e.g. bulk RNA sequencing (10-40%), count-based distributions generally need to incorporate a zero-inflation component since the count model itself imposes a dependence between the mean, variance, and proportion of zeros. Once a zero-inflation component is added, the same challenges arise in distinguishing zeros due to low abundance versus true absence, and some prevalence filter must be applied. Modeling technical vs. biological zeros in more detail would be beneficial in future work, but tests of such methods during development showed that they substantially reduced model robustness and flexibility while providing little inferential improvement. Practically, ANCOM-BC2 attempts to distinguish various types of zeros including zeros due to sequencing and zeros due to true absence. Its high degree of coefficient consistency with MaAsLin 3 on real absolute abundance data, where zero handling is influential to the coefficients, suggests that it almost always considers zeros as coming from true absence, in agreement with MaAsLin 3 (**Fig. 3B-C**).

R1.2.3 *Benchmarks and Comparisons: A direct comparison to DESeq2 and edgeR, which are well-established differential abundance methods, would be valuable. Demonstrating performance differences in the context of read depth and pseudocount sensitivity would provide a clearer understanding of the advantages of this methodology.*

We agree that evaluating a range of differential abundance tools is valuable for informing users about the advantages and disadvantages of each. While DESeq2 and edgeR are popular for identifying differentially expressed genes in RNA sequencing, previous analyses have found them to not be well-suited for microbiome data^{2,5,8,9}. Nevertheless, we included these tools in all synthetic evaluations, and the results are displayed in the updated **Sup. Figs. 2-8** and **10**. Consistent with previous findings, DESeq2 and edgeR were poorly suited for microbiome data with precision typically at or below 0.5 across a wide range of simulations.

Regarding performance differences in the context of read depth, **Sup. Fig. 10** shows that MaAsLin 3 maintains high performance across a range of read depths and that, unlike other differential abundance tools, its inference improves with more sequencing. Again, MaAsLin 3 does not use pseudocounts, but the sensitivity of other tools to pseudocounts is assessed in the newly added **Sup. Fig. 4**, which evaluates the differential abundance tools with and without zero inflation (i.e.,

biological and sequencing zeros or just sequencing zeros). The following text is added to describe the results:

Next, we evaluated the sensitivity of MaAsLin 3 to assumptions about how zeros are introduced into the dataset. By default, SparseDOSSA 2 uses a zero-inflated model, so zeros represent both biological (true) zeros and sequencing zeros (below limit of detection). This potentially impedes methods that use pseudocounts, which effectively assume all zeros are sequencing zeros. When SparseDOSSA 2 was modified to produce only sequencing zeros, the precision of MaAsLin 3 decreased modestly from 0.97 to 0.82, recall and coefficient correlations improved for all tools, and coefficient bias was mostly unaffected (**Sup. Fig. 4**). However, with only sequencing zeros, zeros constituted 39% of the taxonomic abundance table, much less than when biological zeros were also included (77%). The proportion of zeros found in the real datasets subsequently analyzed (66%, 80%, and 97%) was more consistent with a model involving both sequencing and biological zeros, but even when simulating only technical zeros, MaAsLin 3 maintained higher precision than any other tool with similar recall.

Supplementary Figure 4: MaAsLin 3 maintains high precision regardless of whether zeros are due to sequencing or true absence. MaAsLin 3 and other common DA methods were run on 100 synthetic log-normal datasets from SparseDOSSA 2. The datasets were generated as in Fig. 1C but with 100 samples for all datasets and either no zero inflation (sequencing zeros only) or zero inflation (both sequencing and biological zeros). Each metric was calculated as before. 1 is optimal for all metrics except shrinkage, for which 0 is optimal. Each point represents a simulated dataset.

R1.3.1 Counterintuitive Results. The statement, “When applying a q -value threshold of 0.1, the default MaAsLin3 abundance model achieved higher precision than MaAsLin3 with spike-in normalization,” seems counterintuitive. Spike-ins are generally expected to improve precision by reducing technical noise. Could this be due to suboptimal calibration of the spike-in normalization? Further investigation or clarification is warranted here.

We too found this initially counterintuitive. This phenomenon results from the fact that MaAsLin 3 (no spike-in), by incorporating uncertainty in the median in its significance test, typically prevents small coefficients from being significant when the majority of the effect sizes are moderate or large. MaAsLin 3 with spike-ins does not include this extra uncertainty since it is comparing

against zero, not the median, so it is more permissive of false positives with small effect sizes. Indeed, the average absolute effect size of false positive abundance associations for MaAsLin 3 (spike-in) was 0.75, less than the default MaAsLin 3's 0.88. Still, as shown in **Fig. 2A**, for any given recall level, MaAsLin 3 with spike-in normalization has higher precision. The unintuitive result only arises when a significance level is targeted (i.e., not fixing precision or recall in the curve), since no tool perfectly controls the FDR at the nominal level in these complex simulations. The following text has been updated to describe this effect:

When applying a q-value threshold of 0.1, the default MaAsLin 3 abundance model achieved higher precision than MaAsLin 3 with spike-in normalization (average precision 0.85 versus 0.73 at 1,000 samples) despite the former not requiring experimental modification, at least in settings with relatively few true associations (**Sup. Fig. 10A**). Note that this precision is lower than in the comparison above among DA methods, since here a MaAsLin 3 result was only counted as correct if it also correctly specified whether the association was with prevalence or abundance. This phenomenon results from the fact that MaAsLin 3 without spike-in normalization incorporates uncertainty in the median in its significance test, whereas MaAsLin 3 with spike-in normalization compares the coefficients to zero. When some effects are moderate or large, this uncertainty in the median causes small associations (often false positives) to be rejected. Indeed, the average absolute effect size of false positive abundance associations for MaAsLin 3 (spike-in) was 0.75, less than the default MaAsLin 3's 0.88. As expected, this precision difference was balanced by slightly higher recall for the spike-in normalization strategy (recall 0.75 without versus 0.82 with).

In general, spike-ins are not expected to improve precision by reducing noise; if anything, they can increase noise since the normalization involves dividing by a small relative abundance that might be subject to physical technical error. Instead, spike-ins improve precision by quantifying the absolute abundance of each sample, reducing or eliminating the effects of compositionality that would otherwise lead to false positives. Furthermore, spike-ins are not calibrated per se except insofar as the amount of cells or DNA spiked in should be large enough to avoid error but small enough to not overwhelm the original sample in sequencing. In the simulations, the spike-in was chosen to be 1-10% of the relative abundance, concordant with typical experimental values, and the absolute abundance was recorded exactly (i.e., error only comes from the multinomial read sampling, not the spike-in abundance).

R1.4.1 *ALDEx2 Benchmarking. When benchmarking against ALDEx2, were scale-variant priors employed? Using scale-variant priors could enable a more direct, apples-to-apples comparison with the spike-in normalization applied here. See the recent work (bioRxiv: 2023.10.21.563431v1) for reference.*

We apologize for our incorrect assumption that ALDEx2 scale-variant priors were employed by default. We have updated the ALDEx2 models to include both a version using scale uncertainty (no absolute abundance information) and a version using scale information (transformed spike-in information). All instances of ALDEx2 have been replaced with either the first or both of these versions where relevant (**Fig. 1C, 3A, Sup. Fig. 2-8, 10, 14**). Across almost all scenarios, ALDEx2 with scale uncertainty maintained near-perfect precision but at the cost of almost always yielding

the worst recall. ALDEx2 with scale information was more variable, with deteriorating precision in high power settings (large sample sizes), typically moderate recall, and minimal effect size bias similar to MaAsLin 3 (spike-in) when a majority of features were differentially abundant, consistent with the fact that only these two tools directly incorporate external scale information.

R1.5.1 *Figure 4. The addition of a comparison against existing differential abundance methods would be informative. Specifically, showing that this method identifies biologically meaningful signals that other approaches fail to detect would highlight its utility and innovation.*

We agree that a comparison between results produced on the HMP2 by different methods is valuable to highlight the improved biological plausibility of MaAsLin 3's results. While a paragraph comparing the results produced by different models was already included before, we have expanded this section and adjusted **Sup. Fig. 14** with the new ALDEx2 results:

When the other DA methods were applied to the MetaPhlan 4 profiles, most associations identified by MaAsLin 3 overlapped with those of another method (83% were also identified by MaAsLin 2, 32% by ALDEx2, and 4% by ANCOM-BC2; **Sup. Fig. 14**). This overlap suggests MaAsLin 3's results are robust and driven by true biological phenomena rather than variation in methodology. By contrast, among ALDEx2's associations, 64% were unique to ALDEx2, since 177 of its 253 discovered IBD associations involved diagnosis, not dysbiosis (versus 35 of the 244 for MaAsLin 3). Among ALDEx2's dysbiosis associations, 74 of the 76 were also discovered by MaAsLin 3. Consistent with MaAsLin 3's detection of *D. welbionis* loss in UC and CD dysbiosis, MaAsLin 2 and ALDEx2 both estimated significant depletions of *D. welbionis*. ANCOM-BC2 flagged only 11 associations as significant in the entire dataset, all of which overlapped with MaAsLin 3. These results support the plausibility of the new detail shed on the IBD gut microbiome by MaAsLin 3's richer and more nuanced DA model components.

A**B**
Supplementary Figure 14: Most IBD associations discovered by MaAsLin 3 overlapped with other methods.

The species-level abundances from the HMP2 cohort profiled with MetaPhlAn 3 (A) or MetaPhlAn 4 (B) were regressed in each method using a model that incorporated disease-stratified dysbiosis, disease diagnosis, antibiotic usage, age, read depth, and a per-participant random intercept (or, for ALDEx2, a fixed intercept subsequently removed from analysis). Because of the possibility for false positives identified in the simulations, only significant (q -value < 0.1) coefficients with absolute values greater than 1 were evaluated for their overlap.

Reviewer 2

The analysis of differential abundance (DA) to identify microbial features associated with traits of interest is a fundamental question in microbiome research. In this study, the authors introduce a novel DA tool that applies a modified logistic regression to prevalence data and a linear regression to abundance data, claiming superior performance compared to existing methods. The methodology is clearly articulated, the simulation benchmarks are comprehensive, and the real-world application is well presented. However, I have several questions and suggestions that I hope will further refine the text and enhance the technical aspects of the study.

Comments

R2.1.1 *Handling of Zeros. Separating prevalence modeling from abundance modeling is one of the key innovations presented in this work. The authors aim to avoid the common pitfall of adding pseudo-counts to the data. However, my concern is that this approach implicitly assumes that data sparsity arises primarily from biological factors. In reality, technical factors—such as insufficient sequencing depth—could be a dominant contributor to the observed sparsity. This issue is reflected in the simulation results.*

Using the authors' provided code, I modified the sparseDOSSA2 simulator to output the true absolute abundance (with and without spike-in). Under the simulation settings presented in this manuscript, I compared the proportion of zeros in the true absolute abundance versus the observed abundance. The results indicate that they are nearly identical—over 90% of the zeros in the observed data originate from absolute abundance sparsity rather than under-sampling. This suggests that the observed zeros are largely "true" biological zeros rather than artifacts of under-sampling.

Since this setting directly favors prevalence modeling, I wonder if an alternative simulation benchmark could be conducted to evaluate the method's performance when most zeros result from under-sampling.

Many thanks to the reviewer for providing these comments on the manuscript and for evaluating our simulation procedures in depth. We agree that SparseDOSSA 2 assumes data sparsity arises in substantial part from biological factors, an assumption that would only be verifiable in real microbial communities through extremely deep sequencing of samples or broad-spectrum culturing, both of which are broadly infeasible for cost and labor reasons.

However, multiple lines of secondary evidence support this assumption. First, the sparsity observed in the real absolute datasets we analyzed (66%, 80%, and 97% zeros in the mouse, IBD/PSC, and infant datasets respectively) is much higher than in bulk RNA sequencing (10-40%), where zeros are thought to be largely due to technical factors¹⁰. Second, ANCOM-BC attempts to distinguish various types of zeros including zeros due to sequencing and zeros due to true absence (though without then modeling prevalence). Its high degree of coefficient consistency with MaAsLin 3 on real absolute abundance data, where zero handling is expected to explain much of the correlation, suggests that it is almost always considering zeros as coming

from true absence, in agreement with MaAsLin 3 (**Fig. 3C**). Third, whether a zero is due to underlying biology or sampling only affects analysis insofar as it can cause abundance associations to appear as prevalence associations. If a feature has no abundance association and the phenotype of interest is not correlated with read depth (or read depth is controlled for in the model), there will be no prevalence association. An abundance association that results in sequencing drop-out due to low abundance can manifest as a prevalence association, but this is accounted for with MaAsLin 3's prevalence screen. Finally, prevalence associations could always be interpreted from the weaker perspective of analyzing which features were more or less likely to be *detected*, avoiding the question of whether those features are truly absent.

Nevertheless, to address the concern that the SparseDOSSA 2 simulation could favor prevalence modeling, we performed two additional simulations. In the first, to address the concern that MaAsLin 3 is overly reliant on prevalence associations, we varied the proportion of the associations that were due to abundance rather than prevalence. In the second, to address the concern that sampling, not biological, zeros may be dominant, we removed the zero inflation component of SparseDOSSA 2 so that all zeros were due to sampling drop-out. Neither of these additional evaluations substantially changed either absolute or relative performance, and we have added the following text and **Sup. Fig. 4-5** to describe the results:

Next, we evaluated the sensitivity of MaAsLin 3 to assumptions about how zeros are introduced into the dataset. By default, SparseDOSSA 2 uses a zero-inflated model, so zeros represent both biological (true) zeros and sequencing zeros (below the limit of detection). This potentially impedes methods that use pseudocounts, which effectively assume all zeros are sequencing zeros. When SparseDOSSA 2 was modified to produce only sequencing zeros, the precision of MaAsLin 3 decreased modestly from 0.97 to 0.82, recall and coefficient correlations improved for all tools, and coefficient bias was mostly unaffected (**Sup. Fig. 4**). However, with only sequencing zeros, zeros constituted 39% of the taxonomic abundance table, much less than when biological zeros were also included (77%). The proportion of zeros found in the real datasets subsequently analyzed (66%, 80%, and 97%) was more consistent with a model involving both sequencing and biological zeros, but even when simulating only technical zeros, MaAsLin 3 maintained higher precision than any other tool with similar recall. All methods produced similar results when varying the proportion of associations that were related to abundance rather than prevalence, save that the recall of the MaAsLin tools decreased with fewer prevalence associations (**Sup. Fig. 5**). Still, MaAsLin 3 produced a higher recall than any other tool that controlled the false discovery rate, regardless of the proportion of associations that were due to prevalence.

Supplementary Figure 4: MaAsLin 3 maintains high precision regardless of whether zeros are due to sequencing or true absence. MaAsLin 3 and other common DA methods were run on 100 synthetic log-normal datasets from SparseDOSSA 2. The datasets were generated as in Fig. 1C but with 100 samples for all datasets and either no zero inflation (sequencing zeros only) or zero inflation (both sequencing and biological zeros). Each metric was calculated as before. 1 is optimal for all metrics except shrinkage, for which 0 is optimal. Each point represents a simulated dataset.

Supplementary Figure 5: MaAsLin 3 maintains high precision regardless of what proportion of associations are with abundance rather than prevalence. MaAsLin 3 and other common DA methods were run on 100 synthetic log-normal datasets from SparseDOSSA 2. The datasets were generated as in Fig. 1C but with 100 samples for all datasets and a varying proportion of associations with abundance rather than prevalence. Each metric was calculated as before. 1 is optimal for all metrics except shrinkage, for which 0 is optimal. Each point represents a simulated dataset.

Finally, the ANCOM-BC evaluation strategy did not use strong zero inflation (20% “structural” zeros), and MaAsLin 3 performed as well as ANCOM-BC2 in this setting. Overall, this suggests that high sparsity due to biological zeros and prevalence effects, while more consistent with real data, is not necessary for MaAsLin 3 to achieve high performance.

R2.1.2 *Additionally, I would appreciate the authors’ perspectives on how to distinguish between different types of zeros—either within this study or as a direction for future research.*

To fully distinguish between abundance and prevalence zeros, one would need to jointly consider features' relative abundances, prevalences, and samples' read depths¹¹⁻¹³. For each feature, given the relative abundance of the feature if it was present and the read depth of the sample, a binomial model of read sampling could be used to find the probability that the feature is missed due to limited sequencing depth. Then, given the probability the feature was truly present in the sample, Bayes' rule could be applied to calculate the probability the feature was a biological zero given that it was a zero. Of course, we do not know (1) the "true" relative abundance of the feature if it is not detected or (2) the probability of the feature being truly present, but the parameters determining each of these quantities would be what are optimized in fitting the model. Tools such as ANCOM-BC(2) try to distinguish types of zeros, but they do not then use the zeros to model prevalence (except crudely), limiting the interpretability of these biological zeros. Still, when applied to real, experimentally determined absolute abundance datasets, the high consistency of ANCOM-BC2 and MaAsLin 3 coefficients suggests that ANCOM-BC2 is typically assigning zeros as true biological zeros, since theoretical results predict the coefficients to be perfectly correlated without sparsity or with equivalent treatment of zeros (**Fig. 3C**).

While developing MaAsLin 3, we considered models that would jointly consider sampling and biological zeros. However, we decided against pursuing these, at least in the current work, for multiple reasons. First, a primary advantage of MaAsLin 3 is its flexibility to address many different covariate types (e.g., repeated sampling, ordered categorical data, gene expression, and experimentally estimated absolute abundances) and feature types (e.g., functional genes or pathways, in addition to taxa). The methods for detecting, quantifying, and determining whether these features are nonzero typically do not rely on simple counts, and adding the complexity of jointly modeling abundance and prevalence in all these scenarios would have been infeasible.

In particular, almost all current MaAsLin 3 models provide both p-values and standard errors through relatively straightforward means, e.g. t-tests and Wald-type asymptotic z-tests for coefficient values. By contrast, joint modeling would necessitate more general but less interpretable tests such as likelihood ratio tests, which could also be less robust to model misspecification. Second, in the limited scenarios in which we did evaluate such joint modeling during development (the varying sample size analysis), it seemed to provide little advantage over separate abundance and prevalence modeling. Third, the Bayesian calculations described above are straightforward enough for count-based 16S sequencing (i.e., a binomial read sampling model), but for shotgun metagenomic profiling in which abundance does not correspond to individual reads, the problem is much more complicated and case-specific.

R2.2.1 *Effect Sizes of the Abundance Data. This comment builds upon my previous point regarding the handling of zeros. Based on the Methods section and the provided code, the true effect sizes are defined as the log fold changes of the "true" nonzero abundance. This approach may put competing methods at a disadvantage, as they do not explicitly differentiate between different types of zeros. This is particularly relevant for methods that incorporate pseudo-counts, where zeros are treated as technical zeros to some extent.*

I wonder whether this distinction contributes to the poorer performance of competing

methods—such as ALDEx2—in terms of relative shrinkage error and effect size correlation. It may also impact their precision and recall. Again, introducing an alternative benchmark that explicitly accounts for technical zeros could provide additional insights into the performance of different methods.

We agree that using true biological zeros and analyzing effect sizes based on the changes in the non-zero abundance of features can result in poor performance from tools that use pseudocounts such as ALDEx2 and MaAsLin 2. Since MaAsLin 3 and ANCOM-BC2 do not rely on pseudocounts, their effect estimates are typically less biased in these evaluations. Notably though, ANCOM-BC2 still produces worse precision and recall despite modeling different types of zeros.

We newly address this concern by including **Sup. Fig. 4** (above) which shows that, except for sign flipping in shrinkage error by DESeq2 (27% inflation to 19% shrinkage), the shrinkage error is similar with and without zero inflation (<16 percentage point change) for all other tools. This is expected since bias can still be induced by (1) the inclusion of a constant pseudocount despite variation in the true abundance of a feature when missed by sequencing and (2) compositional effects.

Regarding effect size correlation, apart from a 0.31 average improvement by ALDEx2, other tools were only modestly improved by removing zero inflation (0.06 to 0.17 improvement on average). Precision decreased from 0.97 to 0.82 for MaAsLin 3, but its precision remained higher than all tools except ALDEx2 in both scenarios. While ALDEx2 maintained near perfect precision in both scenarios, its recall was 0.12 with zero inflation and 0.71 without, lower than all other tools in both scenarios. As before, the zero inflation model is conceptually more consistent with real microbial community structure, but even without zero inflation, MaAsLin 3 maintains performance similar to or better than all other tools.

R2.3.1 *Assumption on the Median Absolute Abundance Coefficient. As the authors acknowledge, the assumption that at least half of the features remain unchanged with respect to a given metadata may not always hold in certain scenarios. However, the current simulation benchmark primarily evaluates effect size bias and correlation. I believe that precision and recall could serve as more informative metrics in these extreme cases. It would be valuable to see if the authors could provide this additional evaluation to further assess the impact of this assumption.*

We agree that evaluating precision and recall is useful in the situation in which a majority of features are changing, since many tools claim to be robust to such situations. Therefore, we expanded the results from **Fig. 3A** to include prevalence and recall. MaAsLin 3 maintained dramatically higher precision, recall, or (typically) both relative to all other tools by these measures. The following text and figure are updated:

As expected, only MaAsLin 3 and ALDEx2 with spike-in (i.e. experimental) information were fully robust to the assumption violation. By contrast, the correlation between the true abundance

effects and the estimated effects was mostly unaffected by the proportion of true associations for all methods. Despite these biases, ALDEx2 (no scale information) and MaAsLin 3 (with and without spike-ins) maintained high precision regardless of what proportion of features was differentially abundant. All tools except ALDEx2 maintained similar recall regardless of the proportion of features changing, and ALDEx2 with no scale information had much lower recall while ALDEx2 with scale information had much higher recall when a large proportion of features were changing.

Figure 3: Properties of absolute abundance data that are identifiable on the relative scale are well-identified by MaAsLin 3. A. All methods show increasing bias but little change in precision or coefficient correlations when relying on relative abundance data in which more features have true positive associations. MaAsLin 3 and other common DA methods were run on 100 synthetic log-normal datasets from SparseDOSSA 2 generated as in Fig. 1C but with 90% of the associations positive and the sample number fixed at 100. The precision is the proportion of assigned associations with q-values below 0.1 that were true associations. The recall is the proportion of true associations that were identified by the differential abundance tools with q-values below 0.1. The effect size bias is the mean of the fit coefficients minus their true coefficients for true associations. The effect size correlation is the Spearman correlation between the fit and true coefficients per metadatum averaged over the metadata. 1 is optimal for all metrics except the effect size bias, for which 0 is optimal. Each point represents a simulated dataset.

R2.4.1 Clarifications on Level-vs-Level Differences. *Since the authors mention “ordered” levels, I wonder whether a trend test function is incorporated into the main method. If not, does the method rely solely on pairwise comparisons? Additionally, how do users address the more severe multiple comparisons issue in this context? Clarification on this aspect would be helpful.*

We apologize for any unintended confusion regarding the ordered level tests. A strict trend test (i.e., imposing monotonic ordering) was implemented during development but was not found to perform reliably. Instead, the levels of ordered categorical variables are compared using contrast tests between consecutive levels, yielding a number of comparisons equal to the number of levels minus one. These effects and p-values are FDR corrected with Benjamini Hochberg as with all other p-values. To clarify this, we have modified the following text:

For testing differences in levels of an ordered covariate, contrast tests are performed between consecutive levels in the fit model with a right hand side corresponding to zero or the median coefficient difference between the relevant levels. This test is performed with the package lmerTest for linear mixed effects models and with the package multcomp¹⁴ otherwise. No monotonic ordering is imposed in the model per se. The number of tests performed is equal to

the number of levels minus one, and the p-values are FDR corrected with the p-values for all other coefficients.

References

- 1 Casella, G. & Berger, R. *Statistical Inference*. (CRC Press, 2024).
- 2 Nearing, J. T. *et al.* Microbiome differential abundance methods produce different results across 38 datasets. *Nat Commun* **13**, 342 (2022). <https://doi.org/10.1038/s41467-022-28034-z>
- 3 Lin, H. & Peddada, S. D. Analysis of compositions of microbiomes with bias correction. *Nat Commun* **11**, 3514 (2020). <https://doi.org/10.1038/s41467-020-17041-7>
- 4 Zhang, Y., Thompson, K. N., Huttenhower, C. & Franzosa, E. A. Statistical approaches for differential expression analysis in metatranscriptomics. *Bioinformatics (Oxford, England)* **37**, i34-i41 (2021). <https://doi.org/10.1093/bioinformatics/btab327>
- 5 Mallick, H. *et al.* Multivariable association discovery in population-scale meta-omics studies. *PLoS Comput Biol* **17**, e1009442 (2021). <https://doi.org/10.1371/journal.pcbi.1009442>
- 6 Xu, L., Paterson, A. D., Turpin, W. & Xu, W. Assessment and Selection of Competing Models for Zero-Inflated Microbiome Data. *PloS One* **10**, e0129606 (2015). <https://doi.org/10.1371/journal.pone.0129606>
- 7 Hawinkel, S., Rayner, J. C. W., Bijmans, L. & Thas, O. Sequence count data are poorly fit by the negative binomial distribution. *PloS One* **15**, e0224909 (2020). <https://doi.org/10.1371/journal.pone.0224909>
- 8 Hawinkel, S., Mattiello, F., Bijmans, L. & Thas, O. A broken promise: microbiome differential abundance methods do not control the false discovery rate. *Brief Bioinform* **20**, 210-221 (2019). <https://doi.org/10.1093/bib/bbx104>
- 9 Davis, M. L., Huang, Y. & Wang, K. Rank normalization empowers a t-test for microbiome differential abundance analysis while controlling for false discoveries. *Brief Bioinform* **22**, bbab059 (2021). <https://doi.org/10.1093/bib/bbab059>
- 10 Jiang, R., Sun, T., Song, D. & Li, J. J. Statistics or biology: the zero-inflation controversy about scRNA-seq data. *Genome Biology* **23**, 31 (2022). <https://doi.org/10.1186/s13059-022-02601-5>
- 11 Silverman, J. D., Roche, K., Mukherjee, S. & David, L. A. Naught all zeros in sequence count data are the same. *Comput Struct Biotechnol J* **18**, 2789-2798 (2020). <https://doi.org/10.1016/j.csbj.2020.09.014>
- 12 Miao, Z. *et al.* scRecover: Discriminating True and False Zeros in Single-Cell RNA-Seq Data for Imputation. *Statistics in Medicine* **44**, e10334 (2025). <https://doi.org/10.1002/sim.10334>
- 13 Linderman, G. C. *et al.* Zero-preserving imputation of single-cell RNA-seq data. *Nature Communications* **13**, 192 (2022). <https://doi.org/10.1038/s41467-021-27729-z>
- 14 Hothorn, T., Bretz, F. & Westfall, P. Simultaneous inference in general parametric models. *Biom J* **50**, 346-363 (2008). <https://doi.org/10.1002/bimj.200810425>

Dear Editor,

We are very grateful to you and to both referees for the thoughtful feedback on this manuscript, and for your suggestions for improving it during review. We have carefully considered all comments and revised the manuscript with appropriate changes. Detailed responses to each point raised by the reviewers are included here, along with corresponding excerpts from the revised manuscript. We have attached two versions of the revised manuscript, one with modified sections highlighted and one without markup.

Sincerely,

Curtis Huttenhower

Professor, Departments of Biostatistics and Immunology and Infectious Diseases

Harvard Chan Microbiome in Public Health Center

Harvard T.H. Chan School of Public Health

Associate Member, Broad Institute of MIT and Harvard

Reviewer 2

Comments

1. *I appreciate the authors' in-depth explanation regarding zeros arising from both technical and biological factors. This response is informative and, in my view, would strengthen the manuscript and benefit readers if included in the Discussion section. However, as the authors rightly note, the assumption that "data sparsity arises primarily from biological factors" is often unverifiable. It would be appropriate to either acknowledge this as a limitation in the Discussion or clearly state it as a key modeling assumption in the Methods section.*

Many thanks to the reviewer for the time taken to provide these comments. We have included the following sentence in the Methods:

For tractability, all zeros are treated as true feature absence, though such zeros could also arise from technical factors.

We leave an explanation of the assumption to the discussion, where we previously wrote:

Second, limited read depth prevented MaAsLin 3 from always distinguishing correctly between abundance and prevalence associations and sometimes caused it to miss associations. When a rare feature's abundance is associated with a covariate, that feature might be more likely to drop below the limit of detection depending on the covariate's value, yielding an entirely missed effect or a spurious prevalence effect in place of a true abundance effect. Rigorously accounting for this phenomenon would require experimental techniques such as culture-enriched molecular profiling or highly tailored statistical techniques that, for example, distinguish the effects of read depth on the prevalence of taxa, genes, pathways, and other features. In the absence of such methods, MaAsLin 3 reports prevalence associations deemed likely to be spurious and provides diagnostic plots for manual curation when necessary.

2. *I am not sure if I fully agree with the statement, in the rebuttal letter, that "ANCOM-BC2 does not rely on pseudo-counts." According to the user manual, after classifying zero types, differential abundance analysis is performed on taxa not deemed to contain structural zeros—i.e., those whose sparsity is attributed to technical factors such as under-sampling. Pseudo-counts are then added to these technical zeros during a sensitivity analysis. This implies that, once a taxon passes structural zero screening, the presence or absence of zeros still influences significance determination. This deviates from MaAsLin3's assumption that "only the abundances in samples with the feature present are used to determine the abundance p-value." This distinction may help explain why, in Subfigure 4, ANCOM-BC2 showed improved precision (since most zeros are now technical) but still yielded some false positives. I recommend clarifying this in the Discussion or Methods section to better guide users on the underlying assumptions of these methods.*

We apologize for not making a more nuanced statement in the rebuttal letter and for any confusion it caused. While not completely free of pseudo-counts, ANCOM-BC2 relies on pseudo-counts to a lesser extent than all other tools evaluated here besides MaAsLin 3. By using a sensitivity screen for pseudo-counts of various sizes and comparing the results to a model fit with zeros completely removed (i.e., no pseudo-counts), ANCOM-BC2 makes substantially fewer assumptions about the pseudo-counts than ALDEx2 or MaAsLin 2. Furthermore, the correlations in **Fig. 3C** are largely driven by how sparsity is handled, and the fact that the ANCOM-BC2 results agree with the MaAsLin 3 results much more than any other tool suggests that ANCOM-BC2, like MaAsLin 3, is minimally impacted by pseudo-counts. This second point is reflected in the results:

Except for in cases of heavy sparsity, a regression of the relative abundance coefficients on the absolute abundance coefficients should give a slope of 1 and a high correlation. Indeed, for all datasets except the infant gut cohort, MaAsLin 3 and ANCOM-BC2 achieved these results (**Fig. 3**). By contrast, both ALDEx2 and MaAsLin 2—heavily reliant on pseudo-counts that can bias the abundance relationship—produced coefficients with much weaker correlations (average correlation for non-infant datasets 0.52 and 0.59 versus 0.95 and 0.96 for ANCOM-BC2 and MaAsLin 3) and more attenuated slopes (average slope for non-infant datasets 0.46 and 0.45 versus 0.97 and 0.98).

- 3. As a final major point, I encourage the authors to consider including a false discovery rate benchmark using real datasets. One approach would be to generate null data by permuting covariates or applying a similar randomization procedure, so that all findings would, by construction, represent false discoveries. The authors have already implemented this for MaAsLin3; extending it to include other methods would provide a more empirical and assumption-light benchmark for comparing FDR control.*

We have implemented this for MaAsLin 3 in **Sup. Fig. 9**, and the same analysis with the same datasets was performed on many differential abundance methods previously (Nearing et al. 2022). In the text, we note:

These results [for MaAsLin 3] are similar to the corresponding previous analysis for ALDEx2, ANCOM-II, and MaAsLin 2.

- 4. The statement that “ANCOM-BC simulations are somewhat pathological with respect to both sparsity and compositionality” is rather strong. It would be helpful if the authors could elaborate on this characterization or clarify what specifically makes the simulations “pathological.”*

To clarify why this evaluation is particularly challenging, we have added a comment in the methods:

These evaluations are somewhat pathological insofar as a naive logistic regression fit on the prevalence data would suffer from linear separability, and the microbial loads of the two conditions are intentionally very unbalanced.

5. *Optional suggestion: Some studies (e.g., <https://link.springer.com/article/10.1186/s40168-024-01822-z>) suggest that the earlier version of ANCOM-BC, or ANCOM-BC2 applied to data with pre-added pseudo-counts (e.g., +1), may achieve better FDR control under certain scenarios. If deemed useful, the authors might consider including ANCOM-BC or ANCOM-BC2 with pre-added pseudo-counts as an additional comparator in the benchmarking analysis.*

To limit the complexity of our evaluations and evaluate the methods in the way most users would use them, we believe it is most appropriate to use ANCOM-BC2 in its default mode. Already, ANCOM-BC2 handles zeros when identifying structural zeros and determining whether to add pseudocounts based on a sensitivity analysis, so pre-adding a pseudo-count would override these modeling components entirely.